# Flavodiiron-mediated $O_2$ photoreduction at photosystem I acceptor-side provides photoprotection to conifer thylakoids in early spring

Pushan Bag [1,6,7], Tatyana Shutova[1,7], Dmitry Shevela [2], Jenna Lihavainen [1], Sanchali Nanda [1], Alexander G. Ivanov [3,4], Johannes Messinger [2,5] & Stefan Jansson [1] ✉

Green organisms evolve oxygen ($O_2$) via photosynthesis and consume it by respiration. Generally, net $O_2$ consumption only becomes dominant when photosynthesis is suppressed at night. Here, we show that green thylakoid membranes of Scots pine (*Pinus sylvestris L*) and Norway spruce (*Picea abies*) needles display strong $O_2$ consumption even in the presence of light when extremely low temperatures coincide with high solar irradiation during early spring (ES). By employing different electron transport chain inhibitors, we show that this unusual light-induced $O_2$ consumption occurs around photosystem (PS) I and correlates with higher abundance of flavodiiron (Flv) A protein in ES thylakoids. With P700 absorption changes, we demonstrate that electron scavenging from the acceptor-side of PSI via $O_2$ photoreduction is a major alternative pathway in ES. This photoprotection mechanism in vascular plants indicates that conifers have developed an adaptive evolution trajectory for growing in harsh environments.

$O_2$ in the Earth's atmosphere is generated by photosynthetic organisms growing in water and on land. Boreal forests cover 14% of Earth's land (1.9 billion hectares) and account for 33% of Earth's total forests, thereby contributing significantly to the global carbon balance and $O_2$ production[1]. Photosynthetic $O_2$ evolution from $H_2O$ splitting is carried out by a penta-μ-oxo bridged tetra-manganese calcium cluster ($Mn_4CaO_5$) in the oxygen-evolving complex (OEC) of photosystem (PS) II[2] during the light reactions. The electrons extracted from $H_2O$ is further transferred to PSI through several redox carriers and subsequently accepted by $NADP^+$ to produce NADPH[3] in the photosynthetic electron transfer chain (PETC). Later, in the so-called dark reaction, $CO_2$ assimilation occurs involving NADPH in the Calvin-Benson-Bassham (CBB) cycle[4]. The redox imbalance between light and dark reactions often leads to reactive oxygen species (ROS) formation, which can damage photosystems[5]. Hence, several protection mechanisms[6], such as non-photochemical quenching (NPQ) in PSII[7], PTOX-mediated oxidation (chlororespiration) of the plastoquinone (PQ)[8], cyclic electron flow (CEF)[9,10], Mehler-reaction[11] around PSI and photorespiration via RuBisCO in chloroplast stroma[12], have evolved in plants. PTOX, a chloroplastic non-heme diiron quinol oxidase, oxidizes the PQ pool via consumption of $O_2$[13], whereas in CEF, the low pH generated across the thylakoid membrane enhances NPQ[14] and controls the excess electron flow towards PSI[15]. The Mehler-reaction consumes electrons from PSI by reducing $O_2$ to $H_2O$ with $H_2O_2$ (hydrogen

[1]Umeå Plant Science Centre, Department of Plant Physiology, Umeå University, Umeå, Sweden. [2]Department of Chemistry, Chemical Biological Centre, Umeå University, Umeå, Sweden. [3]Department of Biology, University of Western Ontario, London, ON, Canada. [4]Institute of Biophysics and Biomedical Engineering, Bulgarian Academy of Sciences, Sofia, Bulgaria. [5]Department of Chemistry—Ångström laboratory, Uppsala University, Uppsala, Sweden. [6]Present address: Section of Molecular Plant Biology, Department of Biology, University of Oxford, Oxford, UK. [7]These authors contributed equally: Pushan Bag, Tatyana Shutova. ✉e-mail: stefan.jansson@umu.se

peroxide) as intermediate[11], thereby protect PSI. In photorespiration, RuBisCO fixes $O_2$ instead of $CO_2$ and releases $CO_2$ through an inter-mitochondrial and inter-peroxisomal shuttle[12]. Cyanobacteria[16], algae[17], and mosses[18] have an additional flavodiiron (Flv) protein-mediated pathway utilizing excess electrons to reduce $O_2$ directly to $H_2O$ at the acceptor-side of PSI. More recently, a similar mechanism was also predicted to exist in gymnosperms (including conifers)[19–21], but never experimentally verified.

In boreal forests, most plant species overwinter without exposing their green parts to the light, however, conifers needles face extremely high oxidative stress in early spring when solar radiation is high but photosynthesis is constrained by low temperatures[5]. We recently demonstrated the molecular basis of the 'sustained NPQ' mechanism[22] that protects PSII in winter/ES in Scots pine[23] and Norway spruce[24]. Even though this quenching is extremely efficient in protecting PSII, it is hard to completely prevent light-driven ROS production under such low temperatures[25]. Therefore, questions remain about the protection mechanism of PSI. Previous reports have suggested that PTOX[13,26] and CEF may be involved[27]. However, the absence of thylakoid NDH complexes (NADH dehydrogenase-like)[28] in gymnosperms and the potential presence of Flv proteins[19,20] makes the situation complex. Moreover, photorespiration has been shown not to be the major electron sink under low temperatures[29,30]. Obviously, conclusions made from studies of angiosperms may not hold true for conifers[31].

In the present study, we measured light-induced $O_2$ exchange in isolated thylakoids from summer (S) and ES in Scots pine and Norway spruce needles. Using different PETC inhibitors, such as DCMU{3-(3,4-dichlorophenyl)−1,1-dimethylurea} (blocks $Q_B$ site at PSII)[32] and mercuric chloride ($HgCl_2$; profoundly affects electron transfer via plastocyanin)[33], we obtained direct evidence that $O_2$ photoreduction around PSI is much stronger than PSII-related $O_2$ evolution, as PSII remained extremely quenched and PSI activity was higher than PSII in early spring[23]. In combination with P700 absorbance and immunodetection, we demonstrate that Flv-dependent $O_2$ consumption is the major functional electron sink alleviating the over-reduction of the PQ pool by stromal metabolic reductants in early spring when plants are exposed to the combined stresses of cold temperatures and high irradiance. This mechanism could remove internal $O_2$, prevent over-reduction of the acceptor side of PSI, and may protect both photosystems against photooxidative stress in early spring better than other suggested pathways. Our study provides functional evidence of this kind of photoreduction of $O_2$ around PSI and its seasonal variations in vascular plants.

## Results
### Distinct dynamics of $O_2$ evolution in thylakoid membranes of conifers and angiosperms
Isolated thylakoid membranes are devoid of stromal components, their illumination rapidly leads to over-reduction of the PQ pool, which lowers the rate of $H_2O$ splitting. Hence, to measure the "pure" $O_2$-evolving activity of PSII without the influence of other components of the PETC[34], we performed $O_2$ evolution measurements in the presence of PPBQ and FeCy, which accept electrons from the $Q_B$ site on the electron-acceptor-side of PSII (Supplementary Figs. 1 and 2). In intact thylakoid preparations, luminal acidification can quickly suppress $O_2$ evolution, hence, uncouplers need to be employed to dissipate the pH gradient under illumination. This was not required here since we performed experiments with frozen thylakoids that are leaky to protons. First, we measured $O_2$ exchange with a Clark-type oxygen electrode ('Clark-electrode' hereafter) (Supplementary Fig. 1a, b). Second, we performed time-resolved membrane-inlet mass spectrometry (TR-MIMS) assays with 10% of $^{18}O$-enriched water ($H_2{}^{18}O$) (Supplementary Fig. 2) to discriminate between oxygen production (mainly $^{16}O^{18}O$) and consumption (mainly as $^{16}O_2$) reactions, monitored at $m/z$ 34 and $m/z$ 32 signals, respectively[34].

Clark-electrode $O_2$ measurements showed that in spinach thylakoids, the $O_2$ yield was independent of the $O_2$ level in the medium as $O_2$ exchange was similar in both air-saturated and $O_2$-free buffer (Fig. 1a). In contrast, $O_2$ evolution in S pine thylakoid membranes was strongly dependent on the $O_2$ level in the medium (Fig. 1b, c green line). $O_2$ evolution of S pine thylakoid in air-saturated buffer was approximately half of spinach thylakoids. Moreover, the $O_2$ yield reached a maximum after the first 30 s (at 60th second) and then decreased slowly over the next 30 s (60th to 90th second) of the illumination period (Fig. 1b green line). In $O_2$-free buffer, $O_2$ evolution of S thylakoids increased for the first 50 s and then plateaued for the last 10 s (Fig. 1c green line). This data indicates that, in vitro, the $O_2$ production of the S pine thylakoids is inhibited by a photoinactivation mechanism[35] that is dependent on the $O_2$ level in the sample cuvette. Interestingly, spinach membranes in air-saturated buffer without PPBQ and FeCy supplementation did not show any $O_2$ exchange (Supplementary Fig. 1c), whereas ES and S pine thylakoids showed only $O_2$ consumption (Supplementary Fig. 3).

TR-MIMS data of spinach thylakoids were similar to the Clark-electrode measurements and both $^{16}O_2$ and $^{16,18}O_2$ exhibited similar kinetics (Fig. 1d). As the $^{16,18}O_2$-traces at the beginning of illumination are much less sensitive to $O_2$ reduction (due to the initially low $^{16,18}O_2$ concentration in the measuring buffer), this suggests that in spinach thylakoids $O_2$ reduction is of minor importance. The spinach measurements also demonstrate that sufficient artificial electron acceptors have been added to sustain $O_2$ evolution for 60 s under our experimental conditions. In line with the Clark-electrode measurements, for pine S thylakoids (Fig. 1e, f green line) only a transient $O_2$ production is observed for both oxygen species, even at the reduced $O_2$ levels in the MIMS cell. In addition, the $O_2$ consumption showed a transient behavior, possibly indicating a light-induced effect of PSI (see below).

### $O_2$ consumption dominates in early spring pine thylakoids
As the photosynthetic activity differs significantly in S and ES pine needles[23] we compared the light-dependent $O_2$ exchange between S and ES thylakoid membranes using a Clark-electrode. ES samples in air-saturated buffer showed strong $O_2$ consumption (Fig. 1b, blue line) compared to S. In $O_2$-free buffer, $O_2$ evolution in ES very slowly increased until the end of the illumination period, but the overall amplitude of ES samples was ~40% of the S samples (compare blue and green traces in Fig. 1c). This suggests that although light-dependent $O_2$ evolution was present in ES, light-dependent $O_2$ consumption was much stronger from the beginning of illumination, resulting in net consumption of $O_2$. ES samples in air-saturated buffer without electron acceptors supplementation showed ~2.5 times higher consumption than those from S (Supplementary Fig. 3). To understand the dynamics of simultaneous $O_2$ evolution and consumption from a physiological perspective, we performed TR-MIMS also on ES samples, which showed that $^{16,18}O_2$ evolution in ES was much lower than in S (Fig. 1f). However, unlike in S, $^{16}O_2$ was strongly consumed immediately after illumination and consumption reached a maximum after 40 s of illumination in ES (Fig. 1e). PSII showed $^{16,18}O_2$ evolution immediately after illumination in both ES and S but with a much smaller amplitude in ES.

### Water oxidation is not the main source of electrons for photo-reduction of $O_2$ in ES thylakoids
Earlier studies have suggested that high oxidative stress imposed on conifer needles in ES could lead to photoinhibition of PSII via photodamage[36,37]. Mechanisms of photodamage involve a strong reduction in $O_2$ evolution[38] (Supplementary Fig. 4 and the discussion there in) and suggested to form Mn-depleted PSII reaction centers that could consume $O_2$[32]. Our previous study of ES pine needles suggested minor photoinhibition in PSII[23], but it was far from severe under the conditions tested. In the present study, ES thylakoids evolved ~40% $O_2$ compared to S thylakoids (Fig. 1c). Hence, PSII was not severely

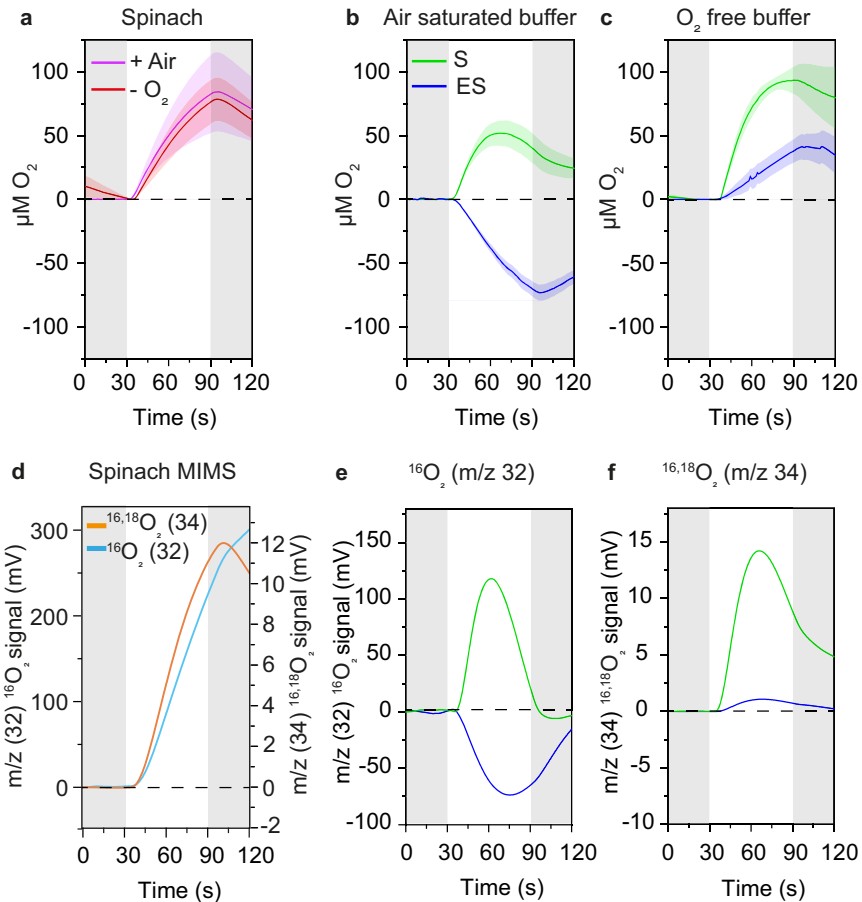

**Fig. 1 | O$_2$ yields in spinach and pine thylakoid membranes. a** O$_2$ yields in spinach thylakoid membranes in air-saturated (+Air) and O$_2$-free buffer (−O$_2$); **b** O$_2$ yields in summer (S) and early spring (ES) pine thylakoid membranes in air, and **c** in O$_2$-free buffer measured with a Clark-type electrode. **d** $^{16,18}$O$_2$ and $^{16}$O$_2$ yields in spinach thylakoid and **e** $^{16,18}$O$_2$ and **f** $^{16}$O$_2$ yields in S and ES pine thylakoid membranes measured with TR-MIMS under partially degassed conditions. Gray-shaded regions indicate dark periods before and after illumination of the thylakoid membranes with 800 (Clark-electrode)/1200 (TR-MIMS) µmol of photons m$^{-2}$ s$^{-1}$ for 60 s (off-sets between start of illumination and O$_2$ production/consumption are intrinsic to the respective technique and do not affect the measurements). In all measurements, thylakoid membranes were supplemented with exogenous electron acceptors, 250 µM PPBQ, and 500 µM FeCy that are sufficient for sustaining electron flow through PSII for 60 s (see spinach data, panels A and D). The transient O$_2$ exchange data for pine thylakoids are thus a consequence of O$_2$ consumption reactions. In spinach, S and ES pine thylakoid membranes data represent O$_2$ exchange corresponding to the same chlorophyll content (50 µg) for both Clark-type electrode and MIMS measurements. For measurements in O$_2$-free buffer, the thylakoid suspension buffer was bubbled with a continuous flow of N$_2$ in the Clark-electrode chamber until the O$_2$ yield became zero prior to the addition of the thylakoid membranes (for the experimental design, see Supplementary Figs. 1 and 2;). Colored shaded regions around O$_2$ yield curve (mean) indicate ±SEM ($n$ = 3) in **a**–**c**, where $n$ = biological replicates. For the TR-MIMS (with H$_2$$^{18}$O enrichment of 10%), one representative trace out of 3–5 independent measurements is shown. Signals processed as described in Supplementary Fig. 2A; note that due to background subtraction 0 mV does not correspond to the absence of O$_2$. O$_2$ exchange rates for the Clark-electrode measurements are given in Supplementary Fig. 11). Source data are provided as a Source Data file.

photodamaged. This makes it unlikely that photodamaged PSII centers could be responsible for the O$_2$ consumption in ES and agrees with an O$_2$ reduction site(s) downstream of PSII in the PETC.

All reduction reactions in the PETC require electrons, which are typically supplied by oxidation of water in PSII. To determine whether electrons involved in the photo-consumption of O$_2$ were supplied from PSII, we added DCMU to the thylakoid suspension. In the presence of DCMU, the $^{16,18}$O$_2$ yield diminished completely, suggesting that H$_2$O oxidation in PSII was completely blocked in both S and ES (Fig. 2b). Surprisingly, both S and ES samples consumed $^{16}$O$_2$, but the negative amplitude of $^{16}$O$_2$ in ES was stronger than S (Fig. 2a). This showed that H$_2$O oxidation in PSII did not supply electrons for the photoreduction of O$_2$ neither in S nor ES. Hence, we conclude that in pine thylakoid membranes, simultaneous O$_2$ evolution and consumption were two independent processes occurring in both S and ES but with different magnitudes. This difference resulted in contrasting net O$_2$ yield patterns between S and ES and the O$_2$ consumption was not PSII dependent.

## Non-photochemically reduced PQ pool supplies electrons to PSI for the photoreduction of O$_2$

PQ, a common biological redox mediator, is present predominantly within the thylakoid lipid bilayer and plastoglobuli[39]. In higher plants, ~30% of the total PQ is photochemically active[40] and the rest is modulated via non-photochemical processes. O$_2$ consumption in the presence of DCMU indicated that the electrons for photoreduction of O$_2$ came from an already stored e$^-$ pool in the thylakoid, which could originate from non-photochemical sources in intact needles in the absence of LEF during early spring[23]. Hence, we first measured the intersystem e$^-$ pool size on intact needles, which suggested that the e$^-$/P700 was 3 times higher in ES compared to S (Fig. 2c). Moreover, post illumination Fo' rise measurements suggested that non-photochemical dark reduction of the PQ pool indeed occurred in ES intact needles (Fig. 2d). This was also confirmed by comparing the changes in the P700 re-reduction upon switching off FR-light illumination, in intact needles with thylakoid samples (Supplementary Figs. 5a–d). To decipher if our isolated thylakoid samples also had an increased

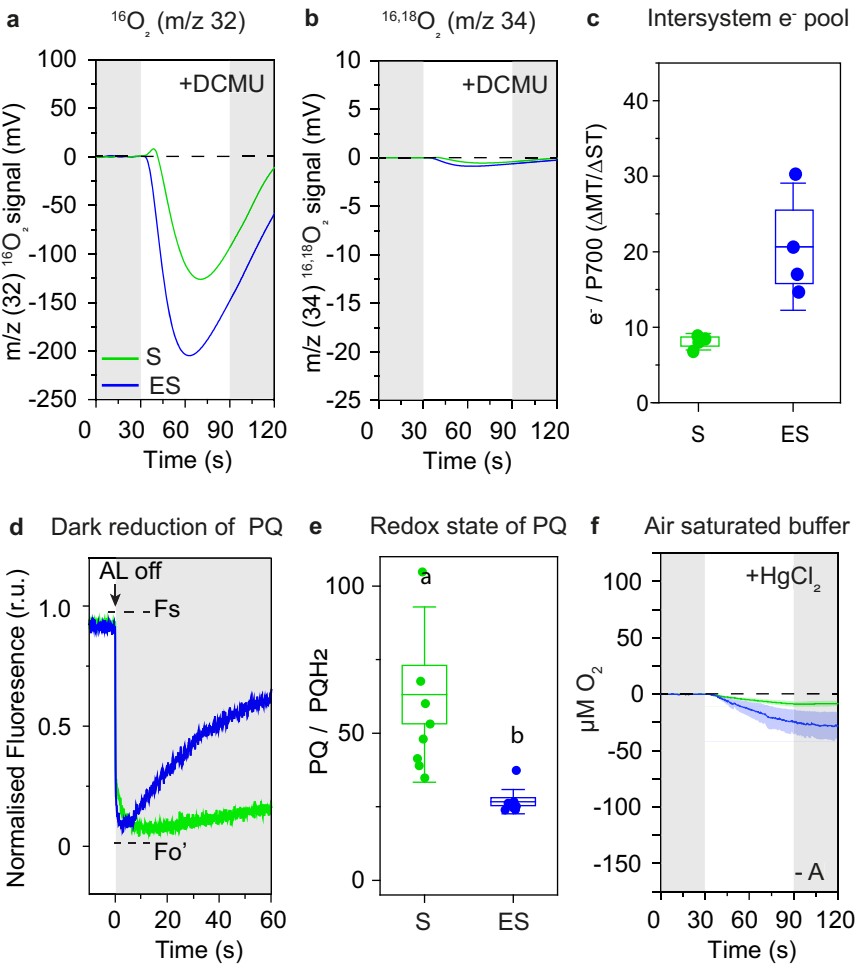

**Fig. 2 | Electron donation for O₂ photoreduction in early spring thylakoid membranes. a** $^{16}O_2$ yields **b** $^{16,18}O_2$ (note that $O_2$ consumption in the mixed labeled $O_2$ was small due to the extremely low natural abundance of $^{16,18}O_2$ compared to $^{16}O_2$) in S and ES pine thylakoid membranes supplemented with 25 μM DCMU measured by TR-MIMS under partially degassed conditions in the presence of 250 μM PPBQ and 500 μM FeCy. Gray-shaded regions indicate dark periods before and after illumination of the thylakoid membranes with 1200 μmol of photons m⁻² s⁻¹ for 60 s. For TR-MIMS (10% $H_2^{18}O$ enrichment), one representative spectrum out of 3–5 independent measurements is shown. **c** Intersystem e⁻/P700 pool measured as a ratio between area under MT and area under ST flash, applied under 200 μmol of photons m⁻² s⁻¹ constant FR-light (720 nm) illumination (n = 3) from S and ES intact needles. **d** Chlorophyll fluorescent signal recorded from S (n = 2) and ES (n = 3) intact needles under constant actinic red light (320 μmol of photons m⁻² s⁻¹) illumination for 5–6 min followed by 120–180 s of dark relaxation (Time scale is set to '0' upon actinic light off). **c, d** n = number of measurements where needles were pooled from 5 trees. **e** Redox state of the PQ pool presented as the ratio of plastoquinol (PQH2) to plastoquinone (PQ) (n = 9) from dark-adapted S and ES thylakoids where n defines independent measurement replication (n = 3) of individual biological replicates (n = 3). Statistically significant mean differences were calculated t test (p < 0.05) (Supplementary Table 5). **f** $O_2$ yields in S and ES thylakoid membranes in air-saturated buffer measured by a Clark-electrode with 1 mg ml⁻¹ $HgCl_2$ supplementation in the absence of PPBQ and μM FeCy (-A). The colored shaded regions around $O_2$ yield curve (mean) indicates ±SEM (n = 3) where n = biological replicates. Data represents $O_2$ exchange corresponding to the same chlorophyll content (50 μg) for both Clark-type electrode and MIMS measurements. $O_2$ exchange rates from Clarke-electrode are given in Supplementary Fig. 11. **c, e** The box bounds indicate ±SEM, minima/maxima indicates ±SD, middle line in the box indicates mean and the dots indicate data points. Source data are provided as a Source Data file.

non-photochemical electron pool, we analyzed the prenylquinones by LC/QTOF mass spectrometry. In dark-adapted thylakoids, the PQ pool in ES thylakoids was predominantly reduced: the PQ/PQH₂ ratio in ES was ~40% lower than in S (Fig. 2e, Supplementary Table 5). In addition, the ubiquinone pool (UQ) was heavily reduced (Supplementary Fig. 5g). Interestingly, the stromal e⁻ pool was previously reported to be 5 times higher in ES pine needles than S[41]. Nevertheless, this suggests that reduced PQ in ES thylakoids upon illumination supplied electrons either to PTOX for photoreduction of $O_2$ to $H_2O$[8] or to PSI through PETC. To distinguish between these two pathways, we measured $O_2$ exchange in air-saturated buffer with/without the addition of mercuric chloride ($HgCl_2$). $HgCl_2$ affects electron transfer via plastocyanin (PC) and hinders electron flow from cytochrome $b_6f$ (Cyt $b_6f$) to PSI[33]. PPBQ and FeCy were not added during these measurements as they can take electrons upstream of Cyt $b_6f$, and hence influence the

effect of $HgCl_2$. We found that $O_2$ consumption in both S and ES was diminished (Fig. 2f) compared to without $HgCl_2$ supplementation (Supplementary Fig. 3). Taken together, these results show that $O_2$ consumption in the ES samples was a photoreduction process occurring around PSI with electrons supplied from the non-photochemically reduced PQ pool and PTOX was not the major site of $O_2$ photoreduction.

## O₂ consumption in ES thylakoids is accompanied by higher P700 oxidation and a higher abundance of flavodiiron proteins

$O_2$ consumption around PSI is known to have profound effects on the redox state of PSI[42]. In line with previous reports[23,27], in our study, P700 became more oxidized (Y[ND] increased) and acceptor-side limitation (Y[NA]) decreased in intact ES needles upon increasing irradiance, compared to S samples (Fig. 3a).

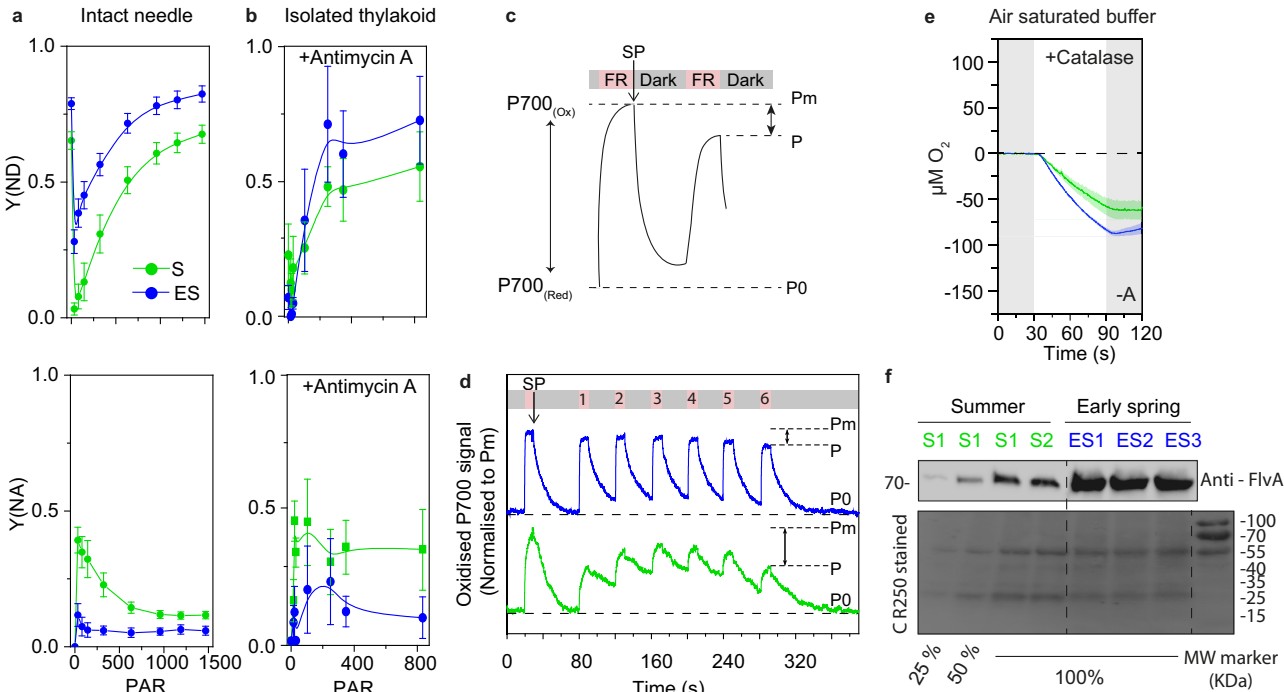

**Fig. 3 | Photoreduction of O₂ around PSI. a** Changes in PSI donor-side limitation [Y(ND)] and acceptor-side limitation [Y(NA)] measured by the SP method with increasing PAR (μmol of photons m⁻² s⁻¹) in S and ES pine needles (n = 3) (without antimycin A), and **b** in S (n = 4) and ES (n = 5) isolated thylakoid membranes (100 μg/ml) (with 30 μM antimycin A). Data in **a**, **b** indicate the mean ± SEM and n = number of measurements where needles/thylakoids were pooled from 3 to 5 biological replicates. **c** Schematic of SP-induced maximum P700 oxidation (Pm) followed by steady-state P700 oxidation (P) achieved by intermittent FR illumination. The effective P700 oxidation state was considered as the difference between the Pm and P signal at the 6th cycle of FR illumination. **d** Changes in P700 absorbance in S and ES pine thylakoid membranes (corresponding to 100 μg/ml chlorophyll) measured with six cycles of intermittent FR illumination in the presence of 30 μM antimycin A (n = 3). **e** O₂ yields in S and ES thylakoid membranes in air-saturated buffer measured by a Clark- electrode with 1000 u ml⁻¹ catalase supplementation in the absence of 250 μM PPBQ and 500 μM FeCy (-A). The colored shaded regions around the O₂ yield curve (mean) indicate ±SEM (n = 3) in the Clark-electrode measurements. S and ES pine thylakoid membranes data represent O₂ exchange corresponds to the same chlorophyll content (50 μg). **d**, **e** n = biological replicates. **f** Relative abundance of flavodiiron A protein in S and ES pine thylakoid membranes. Two summer samples (S1, S2) and three early spring samples (ES1, ES2, ES3) (Supplementary Table 1) corresponding to 4 μg of chlorophyll (100%) were loaded in separate lanes and 1 μg and 2 μg of chlorophyll (25% and 50%) of S1 were loaded in the first two lanes as quality controls, and the membrane was immunoblotted against anti-FlvA antibody. (For the relative quantitation of Flv proteins in S and ES, see Supplementary Table 4). A Coomassie-stained membrane is shown in the bottom panel. Similar immunoblotting results were obtained in four independent experiments. Source data are provided as a Source Data file.

Interestingly, when we compared P700 oxidation (P700⁺) from the fast kinetics of the saturating-pulse (SP) between spinach leaves and S pine needles (Supplementary Fig. 6), we found that in S pine needles, P700 did not show a biphasic re-reduction but instead a transient dip. We interpreted this as follows: P700 first reached maximum oxidation upon the SP trigger, then the oxidation state decreased as electrons were supplied from reduced PQ, but after 100 ms P700 re-oxidized back to maximum as the electrons were consumed from the acceptor side. In ES pine, we detected only a very small transient dip in P700 after 100 ms (Supplementary Fig. 6B) even though e⁻/P700 was higher (Fig. 2c) and PQ/PQH2 was lower (Fig. 2e). This suggests that electrons were taken much faster from the acceptor-side of PSI in ES needles, thereby oxidizing P700. While higher P700 oxidation could in principle be due to CEF[27], recent reports showed that a reduced intersystem PQ pool (Fig. 2c, e) hinders CEF[43]. Therefore, to completely block CEF we used Antimycin A (known CEF blocker)[44]. One earlier report suggested that PGR5 in *Pinus tedea* is resistant to Antimycin A[45] (at 10 μM) when it was expressed in Arabidopsis. However, a very recent report clearly demonstrated that Antimycin A effectively blocks the PGR5-CEF pathway in intact needles of *Pinus sylvestris* (at 200 μM)[27]. Hence, we used 3 times higher Antimycin A concentration than reported previously[45] and at this concentration, Antimycin A was effective in modulating P700 re-reduction kinetics (Supplementary Figs. 5c–f). Nevertheless, we found a similar overall trend in SP measurements, on both thylakoids with Antimycin A and intact needles without Antimycin A, i.e., with increasing irradiance, Y(ND) was higher, and Y(NA) was lower in ES than S (compare Fig. 3a and 3b).

Recently, the SP method was shown not to be fully reliable measure for a possible acceptor-side limitation of PSI as the steady-state concentration of P700⁺ may be influenced by electrons contributed from the PSII via the PQ pool[46,47]. Therefore, to determine steady-state P700⁺, we exposed S and ES thylakoid membranes to six intermittent cycles of far-red (FR) light[48] in the presence of Antimycin A (Fig. 3c). This method does not reflect the maximum oxidized P700 in terms of Y(ND) like the SP method but only indicates steady-state P700⁺ changes with increasing cycles of illumination as a consequence of electron flow within PSI (Fig. 3c). The PSI population in S and ES may be different due to different abundance of PSI[23,24]. Therefore, we normalized the P700 signal to the maximal P700 oxidation level, achieved by the initial SP (Pm) prior to the start of the FR cycles. We found that at the 6th cycle of FR illumination in S samples, the steady-state P700 signal (P) decreased by ~50% compared to Pm, whereas in ES, it only decreased by ~15% (Fig. 3d). Moreover, the relaxation of the P700 signal during the dark periods (after FR illumination) decreased faster and further in ES than in S (also seen in Supplementary Fig. 5). The dark relaxation of P700 is mainly caused by electron donation from reduced PQ. Thus, the results suggested that the non-photochemically reduced PQ in ES donated more electrons to P700 than in S. Overall,

these findings indicated that in ES, a strong electron sink apart from CEF at the acceptor-side of PSI consumes the excess electrons donated to P700 from the reduced PQ.

The Mehler-reaction scavenges electrons from the acceptor-side of PSI consuming $O_2$ and producing $H_2O_2$, subsequently metabolized through the action of (per)oxidases. In isolated thylakoid membranes, stromal components are lacking, and the Mehler-reaction requires external supply of enzymes that could convert $H_2O_2$ to $O_2$ and $H_2O$. To quantify Mehler-reaction-driven $O_2$ consumption in ES thylakoids we performed $O_2$ measurements by the Clark-electrode in air-saturated buffer (without PSII acceptors, PPBQ, and FeCy) supplemented with 1000-unit ml$^{-1}$ catalase. $O_2$ consumption did not change in S thylakoids, but in ES consumption decreased ~34% (Fig. 3e) compared to no catalase addition (Supplementary Fig. 4). This suggested that the Mehler-reaction could explain only a fraction (~1/3rd) of the $O_2$ consumption in ES thylakoid membranes, and that another acceptor at the PSI site accounted for the bigger part (~2/3rd).

As the evidence showed that none of the other possible mechanisms (damaged PSII, PTOX or photorespiration, Mehler-reaction) could explain the major fraction of the $O_2$ consumption in ES thylakoids, we considered the involvement of Flv proteins—as the genes coding for Flv proteins are present in conifer genomes[19]. Spruce FlvA and FlvB are most similar to type 3 Flv proteins in *Physcomitrella* sp. and *Chlamydomonas* sp. (Supplementary Figs. 8b, c), which are known to be associated with photoreduction of $O_2$ at PSI[19,49]. Photoreduction of $O_2$ at the acceptor-side of PSI has recently been demonstrated and suggested to be related to Flv proteins that can scavenge excess electron from PSI and keep P700 in an oxidized state (P700$^+$) in conifers[19]. Therefore, we quantified the abundance of Flv proteins in S and ES pine thylakoids and found that FlvA accumulation was ~3 times higher in ES compared to S (Fig. 3e, Supplementary Table 4) when samples were loaded based on equal chlorophyll amounts.

### $O_2$ photoreduction by Flavodiiron protein is also triggered under artificially simulated semi-early spring conditions in the climate chambers

Thylakoid membranes of pine saplings from climate-chamber acclimated to subzero ($-8\,^\circ$C) temperatures demonstrated stronger $O_2$ consumption compared to control plants (Supplementary Fig. 7a). In presence of HgCl$_2$ most of the $O_2$ consumption disappeared (Supplementary Fig. 7b) but addition of catalase did not change the $O_2$ uptake in any of the samples (Supplementary Fig. 7c), unlike in samples collected from the field in ES (Fig. 3e). P700 oxidation (Supplementary Fig. 7d) and accumulation of FlvA (Supplementary Fig. 7f) were also higher in thylakoid samples from subzero acclimated needles compared to control. This suggests that $O_2$ consumption phenomena in climate-chamber acclimated samples was similar, if not identical, to the natural environment during ES. This indicated that even though in climate chamber the conditions were less harsh than in the field, similar mechanism might have been evoked.

### Flv proteins may mediate consumption of $O_2$ in early spring also in Norway spruce

To elucidate whether photoreduction of $O_2$ at PSI is a general response of conifers to winter conditions, we further explored the possibility of higher $O_2$ consumption and correlation of Flv protein accumulation in Norway spruce thylakoids during ES (Supplementary Figs. 8–10). Norway spruce is a more challenging study system than Scots pine regarding both its biochemistry and physiology. In our study, biochemical preparations of Norway spruce varied more and the "winter states" were less stable during and after preparations than those of Scots pine. However, our data showed that FlvA protein again accumulated in Norway spruce thylakoid membranes, and amounts were ~60% higher in ES compared to S (Supplementary Fig. 8a). Because the accumulation was lower than in pine, the magnitude of changes in

other parameters was expected to be lower. This was indeed the case when ES thylakoid membranes were compared to those in S (Supplementary Fig. 9a): the steady-state P700$^+$ signal decreased by ~30% after the 6th cycle of FR illumination compared to the first cycle. The pattern of fast SP kinetics of P700 oxidation in ES was similar in pine and spruce intact needles (Supplementary Fig. 9b). Clark-electrode measurements with spruce samples did not work in our hands as the thylakoids stuck to the membrane hindering gas exchange. However, in our TR-MIMS assays in the presence of DCMU, ES spruce samples showed stronger $^{16}O_2$ consumption but no change in $^{16,18}O_2$ (Supplementary Fig. 10) compared to S samples. These results suggest that Norway spruce behaved similarly to Scots pine and that Flv proteins were most likely involved in P700 oxidation. However, compared to pine, the magnitude was lower in spruce. This suggests that the Flv-mediated $O_2$ consumption under stressful conditions like in ES is a common phenomenon among certain groups of conifers.

## Discussion

Conifers constitute a large fraction of terrestrial biomass, but in comparison with angiosperms, algae, and cyanobacteria, they are difficult to study mainly because genetic tools are not sufficiently developed. Photosynthesis studies in conifers are also challenging as their photosynthetic tissue—the needles—are significantly different in morphology and chemical composition than that in other plants. However, we recently demonstrated that conifer needles in the winter elicit a protection mechanism[23] that involves direct energy transfer from PSII to PSI, a mechanism whose existence in angiosperms is still a matter of discussion. This protection mechanism may also be associated with phosphorylation of PsbS and triple-phosphorylation of Lhcb1, which trigger thylakoid destacking[24]. Here, we present evidence that an additional protection mechanism that thus far was described in detail only for lower green organisms, namely Flv protein-dependent $O_2$ photoreduction, also contributes significantly to the winter survival of conifer needles by providing protection for PSI. Our study thus provides important experimental support for earlier predictions regarding the involvement of Flv in the photoprotection of gymnosperms[19–21].

Since reverse genetic tools are not available for conifers, we used different PETC inhibitors and measured $O_2$ exchange in thylakoids isolated from S and ES needles. Furthermore, we correlated $O_2$ exchange with protein abundance to understand how conifer needles under some conditions exhibit substantial light-dependent net $O_2$ consumption instead of $O_2$ production as during normal photosynthesis. Four mechanisms—photorespiration, PTOX-mediated chlororespiration, the Mehler-reaction, and Flv pathway—could explain this phenomenon. By using PETC inhibitors, we employed an 'elimination approach' to distinguish between these four possible pathways. Among them, photorespiration in general remains lower in gymnosperms compared to angiosperms, specifically at low temperatures[30]; Moreover, it was easily excluded since we observed $O_2$ consumption in isolated thylakoid membranes. Secondly, Lodgepole Pine in low temperatures has been shown to possess an $O_2$-dependent excess energy dissipation capability that was assigned to PTOX[29]. However, when we added HgCl$_2$ to block electron transfer through plastocyanin, $O_2$ consumption dropped significantly suggesting that the consumption occurred around PSI and not directly from the heavily reduced PQ pool via PTOX (Fig. 2f). Finally, the Mehler-reaction can only take a minor fraction of the total electrons in the PETC when $CO_2$ assimilation is restricted[50] (such as during winter in conifers[27]). Moreover, the Mehler-reaction was previously shown to not be involved in the winter sustainability of conifers[29] and in line with this, experiments with catalase indicated that (Fig. 3e) the contribution of the Mehler-reaction to net $O_2$ consumption in our ES thylakoid membranes was low.

Therefore, the only remaining mechanism that could explain our data is Flv-mediated $O_2$ photoreduction. We acknowledge the fact that we reach this conclusion by eliminating of other possible mechanisms.

A critical proof—demonstration of decreased $O_2$ consumption and photoprotection in mutant lacking Flv proteins—is still not possible to obtain from a conifer; transformation/regeneration protocols and genome editing tools have still not been developed enough. But the fact that genes coding for Flv proteins are among the very few photosynthesis-related genes that are up-, not down-regulated, in Norway spruce in the winter[51,52] which is also reflected in the higher abundance of the protein levels (Fig. 3e, Supplementary Fig. a),—is direct evidence supporting our assumption.

Oxygen reduction via a Mehler-like Flv pathway is known to oxidize P700 by accepting electrons from PSI in cyanobacteria and algae[16,18,48,53,54], and the presence of genes coding for Flv proteins in conifer genomes has been noted by others[19]. The exact site of Flv interaction with PSI remains unclear, but our data suggested that in both examined conifers, FlvA accumulated highly in ES thylakoid membranes (Fig. 3e, Supplementary Fig. 8a). Flv proteins are believed to be soluble proteins, but in pine and spruce, they were retained, at least partially, in conifer thylakoid preparations. Spruce FlvA and FlvB are most similar to type 3 Flv proteins in *Physcomitrella* sp. and *Chlamydomonas* sp. (Supplementary Fig. 8b), which are known to associate with Mehler-like reactions for photoreduction of $O_2$ by PSI[49]. Therefore, our data suggest that Flvs in conifers act as an electron sink for PSI, readily consuming electrons from the acceptor-side during light stress and rapidly reducing $O_2$ to $H_2O$ (Fig. 1c, e, f). In this way, the acceptor-side remains in a sufficiently oxidized state to accept electrons from reduced P700, and P700 upon illumination can readily oxidize by donating electrons to $F_X$ (Fig. 3d, Supplementary Fig. 9a). In addition, $O_2$ consumption would also reduce the risk of ROS production under conditions where the capacity for ROS detoxification mechanisms is low by creating a lower oxygenic environment around the photosystems. Thus, Flv proteins may have a dual protective function. In the winter, and under other severe stress conditions, needle gas exchange is very low due to the thick cuticle and closed stomata. Hence, any $O_2$ produced by $H_2O$ oxidation would accumulate, leading to an increased risk of photooxidative stress. Whether this $O_2$ consumption could give anaerobic conditions in vivo is hard to estimate, but according to our Clark-electrode experiments, 10–15 min of illumination was enough to consume all oxygen in the chamber.

It has been shown that during winter P700 in conifers remain in a donor-side limited condition—meaning P700 is mostly in its oxidized state $(P700^+)$[23,24,27]. However, we found that in ES thylakoids, PQ remains in a highly reduced state (Fig. 2e) concomitant with three times higher intersystem e- pool (Fig. 2c) and 5 times higher stromal e- pool in the intact pine needles[41,55]. In absence or extremely low LEF, this non-photochemical reduction of the PQ (Fig. 2d, Supplementary Fig. 5a–d) might be linked to higher NAD(P)H reduction via chloroplastic NDH II activity as predicted previously[41,51]. How can P700 remain oxidized when the PQ is predominantly reduced? An influx of electrons from the luminal side is passed to the primary stable electron-acceptor $(F_X)$. However, in the presence of Flv activity, electrons could be passed to oxygen, preventing $F_A$, $F_B$ from becoming over-reduced, which would otherwise promote ROS formation[33,56], potentially causing damage[57]. Hence, Flvs could contribute to photoprotection in conifers, in particular in ES, by keeping P700 in an oxidized state to avoid irreversible PSI damage (Fig. 3a–c).

CEF can also lower reduction pressure on PSI by accepting electrons from the PETC[45]. Most conifers lack all plastidial *ndh* genes[58] known to be involved in CEF. We found that P700 remained highly oxidized in ES compared to S (Fig. 3a, c), even when the Prg5/Prgl-CEF pathway was blocked by reduced PQ[43] (Fig. 3d, Supplementary Fig. 9a). Therefore, CEF is unlikely to be a dominant pathway for scavenging electrons and oxidation of P700 in ES thylakoids. However, as PGR5/PGRL1 was reported to be more abundant in ES than S[27], along with higher abundance of ATP in winter acclimated pine needles[55], hence, we speculate that PGR5-CEF may contribute to ATP production via proton gradient formation[59]. In ES when intersystem e- pool is much higher[29,41], over reduction of the PQ could prevent CEF[43] and thereby hinder *pmf* generation[59] which could lower ATP production. However, electron scavenging by Flvs from the acceptor-side of PSI through $O_2$ photoreduction upon illumination could shift the PQ/PQH2 balance and restore CEF. This would maintain the *pmf* and continue ATP generation even if LEF was restricted[23,55].

A schematic of the possible electron flow in ES and S conifer thylakoid membranes is shown in Fig. 4. Upon illumination of S thylakoid membranes, electrons generated from $H_2O$ splitting to $O_2$ reduce the PQ and further pass through Cyt $b_6f$ to PC and then to P700. Upon receiving electrons, P700 becomes reduced and then under illumination becomes re-oxidized by donating electrons to the electron acceptors ($F_X$ to $F_A$ to $F_B$), which are then taken up by Fd for the forward reactions. In ES (**A**), although the $H_2O$ splitting reaction slows down, the PQ pool remains in a highly reduced state (**B**) (most likely with contribution from ndh-2 mediated stromal reduction), which contentiously donates electrons to P700. Upon illumination (**C**), P700 readily becomes oxidized by donating electrons to acceptors ($F_X$, $F_A$ and $F_B$) (**D**). As a result, the acceptors ($F_X$, $F_A$, and $F_B$) become highly reduced as electron demand from the forward reaction is limited due to the down-regulation of the CBB (**E**). However, $F_X$, $F_A$ and $F_B$ remain in an oxidized state as the electrons are taken up by Flv proteins (**F**). Higher Flv activity results in stronger $O_2$ consumption, resulting in net $O_2$ consumption and donor-side limited PSI.

The Flv pathway in cyanobacteria and algae was previously considered 'futile' and consumes a large fraction of electrons from PETC to reduce $O_2$, which may lower the photosynthetic yield[19]. However, it was recently shown that Flvs do not compromise $CO_2$ assimilation[60]. Perhaps Flvs are useful for conifers as a mechanism that can be rapidly invoked when conditions get worse, and dismantled, when they get better. In general, having many parallel mechanisms to protect the precious photosynthetic apparatus would be advantageous, which could be the case for conifers. Although we suggest that Flv-mediated oxygen consumption is substantial in conifer needles in ES, other mechanisms for $O_2$-reduction can be involved in parallel, namely the Mehler-reaction (Fig. 3e). Conifers and angiosperms have evolved from a common ancestor that grew in a suitable but light limited environment[61]. So, why have conifers retained but angiosperms lost the Flv proteins? Under low light conditions, carelessly dissipating the reducing power by routing electrons to $O_2$ would be an evolutionary disadvantage[25]. However, conifers have instead lost the type I NDH-mediated CEF and only retained Pgr5/Pgrl1[58]. It is possible that these different evolutionary trajectories were adaptive as conifers in general adapted to harsh environments but typically competed less well with angiosperms in richer ecosystems. Perhaps Flv proteins provide better protection, but NDH-mediated CEF gives better energy economy under a more favorable environment. Hence, Flvs could be a part of a 'better safe than sorry' evolutionary strategy in conifers.

## Methods

### Plant material harvesting
Fully developed needles (10-15 gm) were harvested 15 times from the south facing branches of 3-5 mature (40+ years old, 8–10-meter tall) trees of *Pinus sylvestris* (Scots pine) and *Picea abies* (Norway spruce) (during 2017-2020 (Sampling dates are provided in Supplementary Table 1). Needles were immediately transferred to the lab (≤5 min) and either subjected to intact needle measurements or thylakoid isolation as described in[62] with slight modifications. After isolation thylakoid membranes were suspended in storage buffer (50 mM HEPES-KOH (pH 7.5), 5 mM $MgCl_2$, 100 mM sorbitol) (previously named as buffer B3 in ref. 62) and flash frozen in 40−50 μl aliquots by liquid $N_2$ and stored in

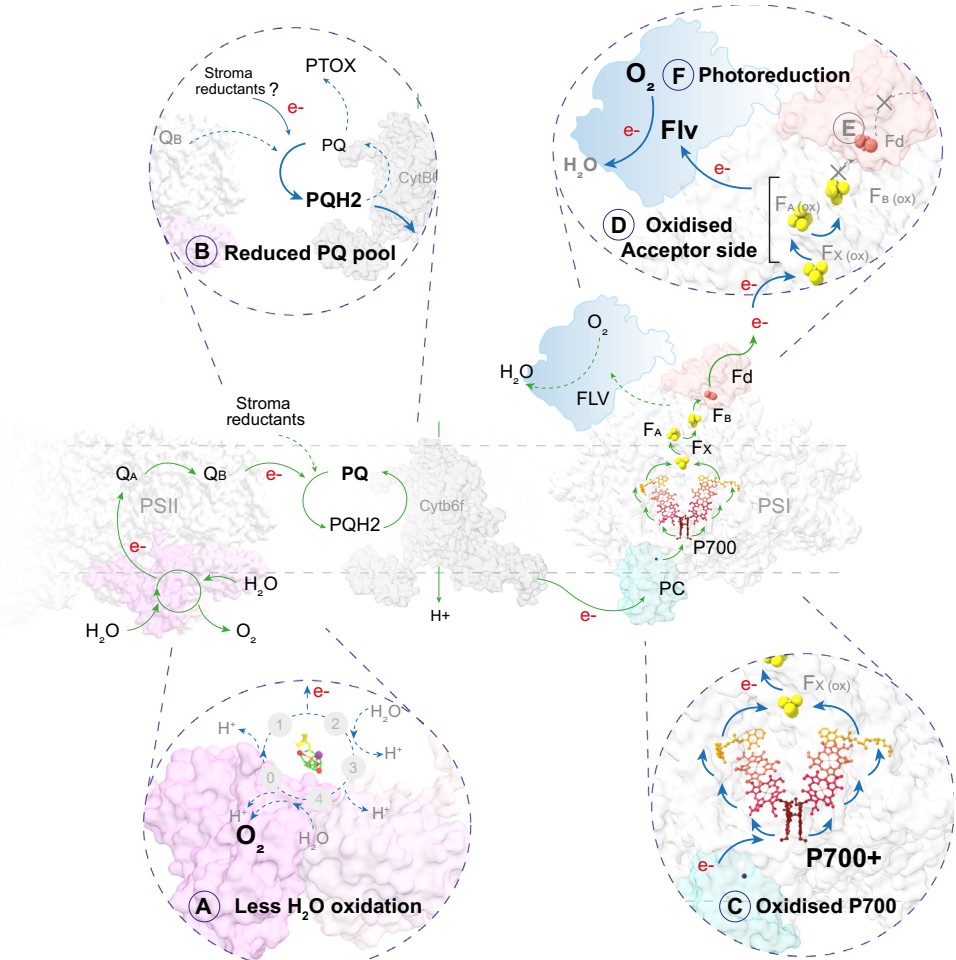

**Fig. 4 | Comparative schematic of light-induced electron flow between summer (green arrows) and early spring (insets with blue arrows) thylakoid membranes.** Insets show possible alterations in the electron flow in early spring compared to summer as directly measured by our experimental setups. Each step of altered electron flow in early spring is indicated sequentially from **A** to **F**. PDB structures used in the model are PSII (*Pisum* sp) 5XNL[79]; Cyt $b_6$f (*Mastigocladus laminosus*) 4HOL[80]; plastocyanin-PSI-ferredoxin (*Pisum* sp) 6YEZ[81]. Note, that the PSII cofactors in contrast to PSI cofactors are not shown in the model as their redox state was not measured experimentally. Electron flow from Cyt b6f to PSI is mediated through plastocyanin that upon binding to Cyt $b_6$f accepts electrons and diffuses to PSI followed by interaction with PSI and electron donation to P700. However, due to the non-availability of Cyt b6f bound plastocyanin PDB structure plastocyanin is only shown in association with PSI. The position of Flv on PSI does not represent any empirical binding site nor define the exact site of electron transfer from PSI acceptor side to Flv. Bold text in the schematic indicates direct experimental evidence of the phenomena.

−80 °C until further use. Before use chlorophyll concentration of each aliquot was determined following[63].

Forclimate-chamber experiments, small 2–3 years old pine saplings (~1 meter tall) were acclimated to control condition with 120 μmol of photons m$^{-2}$ s$^{-1}$ at 18 °C for 10–14 days, and needles were harvested from 12 saplings and subjected thylakoid isolation. Then 15 saplings were moved subzero temperature (−8 °C) and acclimated for 10–12 days and needles were harvested and subjected to thylakoid isolation. In all climate chamber experiments n = number of biological replicate groups, where one group comprised of thylakoids pulled from 4 plants in control and 5 plants in −8 °C conditions, hence, total 12 plants pulled into 3 groups (i.e., n = 3) in control 273 and 15 plants pulled into 3 groups (i.e., n = 3) in −8 °C conditions.

**Clark-type electrode O$_2$ measurements**

Light-induced O$_2$ evolution/consumption was measured in a Clark-type electrode (Hansatech instruments, England) with 800 μmol of photons m$^{-2}$ s$^{-1}$ white LED light for 60 s (see Supplementary Fig. 1 for more details). First background signal was determined by only adding 500 μl of B3 buffer in the Clark-type electrode chamber. Later, for all O$_2$ measurements by the Clark-type electrode, thylakoid membranes

were resuspended (thawed in dark on ice in case of frozen samples for 30 min) in storage buffer supplemented with/without 250 μM PPBQ (2-phenyl-$p$-benzoquinone) and 500 μM FeCy (potassium ferricyanide) as exogenous electron acceptors for PSII[64] to a final volume of 500 μl and measurements were performed immediately (as described in Supplementary Fig. 1A). The temperature was maintained at 10–12 °C during all the measurements. For inhibition of PSII activity, DCMU (3-(3,4-dichlorophenyl)−1,1-dimethylurea) (dissolved in DMSO) was added at a final concentration of 25 μM. For blocking electron transfer through plastocyanin, mercuric chloride (HgCl$_2$) was added at a final concentration of 1 mg ml$^{-1}$. For catalase activity catalase (Sigma) was added at a final concentration of 1000-unit ml$^{-1}$ from a stock concentration of 5000-unit ml$^{-1}$ in water. In case of measurements in O$_2$-free buffer (N$_2$ was bubbled in the Clark-type electrode chamber), first 500 μl of storage buffer supplemented with 250 μM PPBQ and 500 μM FeCy was added in the Clark-type electrode chamber and the chamber was closed. N$_2$ was supplied in the chamber through a needle from a pressure-controlled gas tap. Once the O$_2$ signal reached zero (no O$_2$ left in the chamber), thylakoid samples were added which caused an overshoot in the O$_2$ signal. N$_2$ bubbling was continued for 8–10 s further to bring the O$_2$ signal back to zero. Illumination started

once $O_2$ yield reached Zero (Supplementary Fig. 1B). The data shown in all Clark-type electrode measurements represent total $O_2$ exchange from thylakoid membranes corresponding to 50 µg of chlorophyll content.

## Time-resolved membrane-inlet mass spectrometry (TR-MIMS) measurements

Gas exchange was measured in thylakoid suspension (in Buffer B3) by the TR-MIMS setup as previously described[65]. The MIMS setup contained an isotope ratio mass spectrometer (Delta V $^{Plus}$; Thermo Fischer Scientific) connected to an in-house built gas-tight membrane-inlet chamber (200-µL) and a cooling trap. Prior to the measurements, $H_2^{18}O$ (97%; Larodan Fine Chemicals AB) was added (to a final enrichment of 10%). Analysis of $O_2$ reactions was performed on the $m/z$ 32 ($^{16}O_2$) and $m/z$ 34 ($^{16,18}O_2$) signals, with Faraday cup amplification of $3 \times 10^8$ and $1 \times 10^{11}$ (in the present study, the signal $m/z$ 36 ($^{18}O_2$) was not analyzed due to its' low amplitude at the employed $H_2^{18}O$ enrichment and its overlap with the $^{36}Ar$ ($m/z$ 36) signal). Continuous saturating Illumination (1200 µmol of photons $m^{-2} s^{-1}$) was provided by external white light source (S2 High-Intensity White Light Source; Hansatech Instruments Ltd.). The measurements were performed at 20 °C under continuous stirring of the sample suspensions diluted with the storage buffer containing electron acceptors, 250 µM PPBQ and 500 µM FeCy. Recording of the MIMS signals was started immediately after the samples were loaded in the MIMS cell and covered with the plunger. Light was switched after 120-s incubation of the samples in the cell. The final $O_2$ exchange curve was obtained after correction of the gas consumptions by the mass spectrophotometer (Supplementary Fig. 2). The correction was performed by fitting the traces recorded in the dark period before the illumination period using Origin Pro 2021 for each individual mass signal (see Supplementary Fig. 2 for further details of the data analysis). A lag time (of a few seconds) observed between the onset of sample illumination and monitored $O_2$ evolution/consumption represents an offset caused by the instrumental setup (described in ref. 31). Such offset did not affect the monitored differences between ES and S samples.

## P700 and fluorescence measurements

P700 measurements were performed either on intact needles or on thylakoid membranes with saturating-pulse (SP) or on thylakoid membranes with intermittent cycles FR illumination by Dual PAM100 (Walz). First, needles or thylakoid membranes were dark-adapted from 15–20 minutes. For SP measurements on intact needles P700 absorbance was recorded on a FR-light background followed by application of a 600 ms SP of 4000 µmol of photons $m^{-2} s^{-1}$[42]. For SP measurements on thylakoid membranes, P700 absorbance was recorded on a FR-light background followed by application of a 50 ms SP of 1000 µmol of photons $m^{-2} s^{-1}$. For measurements on needles PAR (photosynthetically active radiation) was increased step by step up to 1600 µmol of photons $m^{-2} s^{-1}$ whereas for measurements on thylakoid PAR was increased step by step up to 800 µmol of photons $m^{-2} s^{-1}$.

For intermittent cycles of FR-light measurement on thylakoids, first, maximum P700 oxidation was determined by a 50 ms SP of 1000 µmol of photons $m^{-2} s^{-1}$, followed by 30 s of dark interval, and then P700 signal was monitored for six cycles of 10 s FR illumination (intensity was 250 µmol of photons $m^{-2} s^{-1}$) followed by 30 s of dark interval between each FR illumination[48]. In this measurement thylakoid samples were dark incubated in the presence of 30 µM Antimycin A for blocking Prg5/Prgl1 mediated cyclic electron flow.

The transient reduction of FR-light-induced (200 µmol of photons $m^{-2} s^{-1}$, 720 nm, 5 min) steady-state P700$^+$ signal caused by single turnover (ST) and multiple (MT) saturating flashes of white light was used for estimation of the apparent intersystem electron ($e^-$/P700) pool size in vivo in both S and ES pine needles[66,67]. MT saturating flashes (50 ms) and ST saturating flashes (half peak width 14 ms) were applied by the power/control units of Dual PAM100 (Heinz Walz GmbH, Effeltrich, Germany). The complementary areas between the transient reduction/re-oxidation of P700 after ST and MT applications and the steady-state level of P700$^+$ under FR illumination were used for estimation of the functional pool size of intersystem electrons on a P700 reaction center basis as described in ref. 67. For calculating the area under the curve, the curves were integrated with minimum and maximum P700 signals as the limits by using Origin Pro 2021.

For P700 re-reduction measurement, first P700 oxidation was induced via FR-light for 3 minutes, then FR-light was switched off and the re-reduction rate was calculated in terms of time constant Tau ($s^{-1}$) by using a mono-exponential decay fit as done previously in refs. 68,69. Under FR illumination PSI was preferentially excited, and thereby drew electrons from the PQ pool that were present in the intersystem $e^-$ pool prior to the FR illumination. In addition, FR-light activates PSI-dependent CEF. With time, under FR-light illumination, PSI kept draining electrons from the intersystem $e^-$ pool and reached a steady-state (after 25–45 s). When this measurement was performed on intact needles, where all stromal components are intact, the re-reduction kinetics of P700 after the FR-light is switched off is used as a reliable measure of both PSI-driven CEF and/or electron flow(s) reducing the PQ pool non-photochemically in darkness via different stromal reductant pathways. In the presence of Antimycin A which effectively blocks PGR5-dependent CEF in pine[27], P700 re-reduction would reflect only the electron flow(s) through the combination of different non-photochemical pathways.

In vivo Pulse/Amplitude chlorophyll fluorescence measurements of both S and ES pine needles were performed as described earlier[23]. The reduction state of plastoquinone (PQ) was assessed following the post-illumination increase of chlorophyll fluorescence at the Fo' level[70–72]. Chlorophyll fluorescence of both S and ES pine needles was measured in vivo using Dual PAM100 chlorophyll fluorescence measuring system (Heinz Walz GmbH, Effeltrich, Germany). After 20 minutes of dark adaptation, the intact needles were subjected to a constant actinic red light (AL) of 320 µmol of photons $m^{-2} s^{-1}$ illumination for 5–6 minutes until a steady-state level of Fs was reached as reported previously in intact pine needles[55]. Then the actinic light was switched off and the transients from Fs to Fo' were recorded for 60 s. Post illumination rise of Fo' is not influenced due to induction of artificial quenching by AL light intensity[23]. The fluorescence signals of both S and ES needles were normalized between Fs and Fo' for direct comparison of post-illumination rise of Fo' and the time scale was set to '0' when the AL was switched off.

## SDS-PAGE and immunoblotting

After chlorophyll concentration was determination by ref. 63, thylakoid membranes containing equal amounts of chlorophyll were separated on 4–20% TGX PAGE gels (Bio-Rad Criterion TGX Any kD™ precasted gels (Bio-Rad, #5678124)[23]. Thylakoid membranes samples ($n$ = 3–5) were solubilized in Laemmli sample buffer[73] supplemented with 100 µM DDT. After SDS-PAGE, proteins were electrophoretically transferred on a nitrocellulose membrane (Merck) by wet transfer at 15 V for 6 h at 4 °C and blocked with 2% skimmed milk at room temperature for 2 hours. After blocking, membranes were incubated with specific primary antibody against FlvA (1:2000 dilution, provided by Prof. E.-M. Aro) overnight at 4 °C and then incubated with secondary anti-rabbit antibody (AgriseraAB, Vännäs, Sweden−AS101461 at 1:15000 dilution) for 2 hours at room temperature. The antibody complexes were detected using ECL reagent (#AS16 ECL-S-N, AgriseraAB). Images were captured in Azure imaging system.

## Prenylquinones in thylakoid membranes of pine

Three aliquots of the isolated thylakoid samples (-50 µg of thylakoid in 10 µl) were kept in dark (ambient state, $n$ = 9). Thylakoids were pelleted

down by centrifugation (2700 × *g*), sample buffer was removed and three glass beads (3 mm diameter) and 100 μl of ice-cold 30:70 (chloroform:methanol) containing labeled internal standards (Tocopherol) were immediately added to the samples. Samples were extracted for 5 min in a multi-vortex and centrifuged at 4 °C at 18,000 × *g* for 5 min. Supernatant was transferred into an insert in a vial and samples were analyzed by liquid chromatography-mass spectrometry (LC-MS) with a method adopted from ref. 39. Samples were directly injected in the extraction solvent without drying the aliquots to prevent changes in the redox state of quinones. In addition, quality control (QC) samples were prepared by combining an aliquot of each sample group and treatment. Based on the QC samples that were run between samples sets, the state of quinone pool did not change during the whole analysis period. Furthermore, an aliquot of QC sample was mixed with high concentration of ascorbic acid (in methanol, 50 mM) to artificially reduce the metabolite pools and to identify redox active quinones in the thylakoid samples.

The LC-MS system consisted of Agilent 1290 Infinity LC and 6546 LC/QTOF mass spectrometer equipped with atmospheric pressure chemical ionization (APCI) source and DAD (diode array detector) (Agilent Technologies). Metabolites (1 μl sample) were separated with Acquity UPLC BEH C18 column (Waters, 100 × 2.1 mm, 1.7 μm particle size) combined with an Acquity UPLC BEH C18 VanGuard pre-column (Waters, 5 × 2.1 mm, 1.7 μm particle size. Column temperature was set to 60 °C, autosampler temperature to 10 °C, and flow rate to 0.5 ml/min. Mobile phase (A = water, B = Methanol) composition at the start was 80% B increasing to 100% B in 1 min, maintained at 100% B for 6.5 min followed by a re-equilibration at 80% B for 1.0 min. The mass spectrometer parameters were optimized with α-tocopherol to increase the sensitivity and to detect mainly molecular ions. The samples were run with positive and negative mode. Data were acquired with a scan time of 4 scans/s and with a range of m/z 120–950. The corona current was 4 μA in positive and 16 μA in negative mode and nebulizer was 45 psi in positive and 40 psi in negative mode. Drying gas flow was 6 l/min, gas temperature 300 °C, vaporizer temperature 350 °C and capillary voltage 3000 V. Fragmentor was set to 120 V, skimmer to 65 V, and octopole RF peak to 750 V. In addition, MS/MS spectra were acquired with positive and negative APCI mode for QC samples over a range of collision energy 10, 20, and 40 V. Prenylquinones (11 compounds) were identified based on the standard compounds (α-tocopherol), accurate mass measurements and MS/MS fragmentation patterns (See Supplementary. Table 3). Data were processed with Mass Hunter Profinder (version B.08.00, Agilent Technologies). Prenyllipid levels were expressed relative to the total metabolite pool and the redox state based on the ratio between the oxidized (quinone) and reduced (quinol) forms.

### Statistical analyses

The effects of season variation on quinone redox status (ratio of oxidized and reduced forms) were tested with *t* test (IBM SPSS statistics version 27) (See Supplementary Table 5). Statistically significant differences of FlvA accumulation between samples collected on different dates of summer and early spring were calculated by one-way ANOVA (SPSS) and the effect of season by t-test (SPSS) (See Supplementary Table 4). Pairwise comparisons were performed with Fisher's Least Significant Difference test (LSD) (See Supplementary dataset 2). In all cases, *P* value < 0.05 was considered significant.

### Bioinformatic analyses of FlvA and FlvB in spruce

FlvA and FlvB protein sequences were generated from the FlvA and FlvB gene model obtained from the Congenie database (https://congenie.org/) by using Expasy translate tool (https://web.expasy.org/translate/)[74] and was aligned with other flavoproteins in other species (See Supplementary dataset 1) by using Clustal Omega multiple sequence alignment tool (https://www.ebi.ac.uk/Tools/msa/

clustalo/)[75]. The unrooted phylogenetic tree was constructed by using IQ-TREE 2[76] from the aligned sequences. Spruce FlvA and FlvB domains were predicted by using NCBI CDD search tool (https://www.ncbi.nlm.nih.gov/Structure/cdd/wrpsb.cgi)[77] and Expasy PROSITE tool (https://prosite.expasy.org/)[78].

### Reporting summary

Further information on research design is available in the Nature Portfolio Reporting Summary linked to this article.

## Data availability

All raw and source data for Figs. 1–3 and Supplementary Figs. 1–11 are provided in the source data file. Peptide sequences for phylogenetic analysis are provided in supplementary dataset 1 and statistical details for Supplementary Table 4 are provided in Supplementary dataset 2. All steps of data analysis to reach the source data from the original raw data are provided in the pipeline in the source data file. Source data are provided in this paper.

## Code availability

All codes and functions used for data analysis (baseline correction/signal drift correction/fitting) are provided in the corresponding figure legends.

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

## Acknowledgements

This project was supported by SE2B Horizon 2020 under grant agreement no. 675006 (SE2B) to S.J.; the Swedish Research Council (VR) to S.J. (2016-04894 and 2021-05062) and J.M. (2020-03809); the Kempe Foundation (2014), FORMAS (2015-00907 and 2021-01474), SSF (FFF20-0008), Vinnova (2016-00504) and KAW (2016-0352 and 2020.0240) to S.J. and; KVA (BS2022-0021) and Stiftelsen JC Kempe Memorial Scholarship Fund (2021) to P.B. We would like to thank the photosynthetic platform at UPSC and IRMS platform at KBC (Department of Chemistry, Umeå University) for support with the Clark-electrode and MIMS measurements, respectively and the Swedish Metabolomics Center (SMC) for their support in metabolite analysis. We are also grateful to Professor Eva-Mari Aro and colleagues who gave useful comments and, after testing and confirmation, provided antibodies recognizing conifer Flv proteins.

## Author contributions

P.B., T.S., and S.J. conceived the idea; P.B, T.S., D.S., J.L., A.G.I., J.M., and S.J. designed the research; P.B. and T.S. performed Clarke-electrode measurements; T.S. and D.S. performed TR-MIMS measurements; J.L. performed the prenyl pool measurements; P.B. and A.G.I. performed P700 measurements, P.B. and S.J. performed the bioinformatic analysis; P.B. and S.N. performed immunodetection; J.M. and S.J. contributed to reagents, tools and supervised experiments; S.J., J.M., and P.B. acquired fundings; P.B., T.S., D.S., A.G.I., J.L., J.M., and S.J. analyzed the data; P.B., J.M., and S.J. wrote the paper with input from all co-authors.

## Funding

## Competing interests

The authors declare no competing interests.
