## [Peer Review File · Nature Communications]

Reviewers' Comments:

Reviewer #1:

Remarks to the Author:

Bag et. al present evidence that there is a strong photoprotective effect against damage by high light at low temperatures in scots pine and to some extent also Norway spruce during early spring as compared to summer. This is reflected in a net O₂ consumption rather than an O₂ release. In their manuscript Bag et al interrogate the different points of the electron transport chain to determine where O₂ consumption occurs and ultimately where the electrons end in order to determine the mechanism at play. In the end, the conclusion is that none of the canonical protective mechanisms that typically are at play in plants display this effect. Instead an increased expression of Flv proteins (similar to moss and cyanobacteria) is shown and by inference it is concluded that these act as an electron sink by reducing excess O₂.

The manuscript is clear and well written with a reasonable conclusion, albeit it is not hammered in with 7-inch nails. I have no major comments related to the science presented as it is all very sound. Overall I found the manuscript a great read with some very nice and clear well explained experiments. Therefore I think this manuscript would be a nice addition to the Nat. Comm. portfolio.

I do have a list of minor comments, that should be considered by the authors.

Minor comments

Line 37 – please introduce the shorthand ES already here.

Line 50 to 53 – This may seem a finicky comment, but these four lines had a large influence on my want to continue reading, so I will present the comment regardless of how small it may seem. As a reader, I am confused by the abrupt statement about plankton and immediate transition to boreal forests, where the relevance of these forests for O₂ production is defended. It would be better just to say that boreal forests contribute significantly to O₂ production due to their widespread presence in the northern regions.

Line 125 – Fig 1D seems to be a CTOE experiment not a MIMS experiment.

Figure 1 – Is there a good reason for the hierarchy of labels in this figure? Otherwise I suggest to change the labelling to be a – b – c in the first row and d – e – f in the second row or a – c – e in the first and b – d – f in the second. As it is now is confusing. – also very confusing that m/z 34 is shown to the left of m/z 32 (similar for Figure 2, that 32 is to the right of 34)

Line 331 – In lune should be in line

Line 440/441 – Do you mean to say that the samples contained 25% DMSO or was it perhaps a 100 mM stock

Line 1215 – Provide = provided

References citing Science papers have formatting issues – 3, 4, 83

It might be an active choice, but you seem to have lost all superscripts and subscripts in your references and in some cases even have spaces inserted between H and 2 in H₂ (ref. 5)

Reviewer #2:

Remarks to the Author:

Scots pine and Norway spruce show the strong O₂ consumption in early spring. By eliminating other possibilities, the authors claim that this high O₂ uptake depends on Flv. In fact, the Flv protein level increases in early spring. They observed the strong O₂ uptake even in the presence of DCMU. They suggest that the origin of electrons is the stromal pool, as well as PSII.

I understand that the genetical approach is not feasible in the non-model plants. However, the contribution of Flv to the photoprotection in the plants is not very surprising. The authors could not

directly prove it in the process. Another serious problem is the lack of evidence on the electron donor to the pathway. I guess that they consider the contribution of type II NDH. This is also likely but the direct experimental evidence is missing especially because of the problem in Antimycin A (see below). The manuscript contains some interesting discoveries but I have to say that it lacks the solid conclusion.

Other specific comments

- 1) Figure S1. Yellow is hard to see.
- 2) Briefly explain the essence of TR-MIMS assay in the main text. It would be helpful for readers if you explain that water was labelled.
- 3) The text includes rather many abbreviations. I often had to go back the story to confirm some abbreviations. Is it necessary to use so many abbreviations, CTOE, MR, PR, PCEF, ...?
- 4) Explain green and blue lines in all figures.
- 5) Ls124-130. This paragraph is hard to follow. Fig. 1B (L125)? in ES pine thylakoids (Ls128-129)?
- 6) L178. in the lumen of chloroplasts??? PQ is in the thylakoid membrane and PG. PQ in mitochondria?
- 7) Figure 3B. It was reported that PGR5-dependent CET was resistant to Antimycin A in *Pinus sativa* (Sugimoto et al., 2013). Is it possible to confirm the amino acid sequence in your plants? At least, you have to mention the possibility that CET is resistant against Antimycin A. Without further confirmation, it is difficult to conclude anything on PGR5-dependent CET.
- 8) L231. What are luminal components. Does the plant have Cyt_c6? Do you consider other electron donors to P700?
- 9) L237. Does catalase function in chloroplasts in the plant?
- 10) Figure 3E. Why did not not you combine Figure 3E and S3? 35% of O₂ reduction is via MR in ES plants. You cannot ignore its contribution. In S plants, contribution of MR is really minor.
- 11) L260. Smaller pines? Define the plant materials scientifically. Size, age, ...?
- 12) PCEF via Flv. From this term. I imagine the electron flow from water (generated by PSII) to water. But you suggest that the origin of electrons is mainly NADP. Is this PCET?
- 13) Prg5/Prg1 should be PGR5/PGRL1.
- 14) L389. ndh-2 homolog? Do you mean the different enzyme from type II NDH?
- 15) Ls396-398. What do you mean by primitive? Ref68 reported the exogenous Flv system in rice.
- 16) L407. I was really confused. Conifers do not contain type I NDH (photosynthetic complex I). Does your plant also lack type II NDH?
- 17) There are many typos in the text.

Reviewer #3:

Remarks to the Author:

Bag et al. studied O₂ photoreduction in conifers. The photoreduction of O₂ mediated by Flv has already been reported by several recent literatures (Noridomi et al. 2017; Ilík et al. 2017; Takagi et al. 2017). The main novelties of this work are (1) quantitative evaluation of Flv-mediated O₂

photoreduction at the level of isolated thylakoid membranes and (2) seasonal difference of it. These new findings are worth being published in Nature Communications if they are properly verified. However, there are several concerns as follows:

Line 55-73, Terminology for photosynthetic electron transport is strange. Flv can be categorized into "Mehler-like reaction". MR is also funny abbreviation for this. Pseudo-CEF is an often-used word but I would suggest "Mehler-like reaction" is more suitable to Flv, because "pseudo-CEF" should be difficult to understand for non-expert.

Line 84, I agree with that the activity of photorespiration is lower in gymnosperms than in C3 plants, and further decreases at low temperature. However, certain activities have been measured by previous literatures e.g., (Hanawa et al. 2017), and the authors can mention this more.

Line 88-91, This sentence is strange and should be clarified more. Normally, we cannot imagine that PSI uses the electrons much more than generated at PSII.

Line 103-130, This section needs to be seriously improved. The measurements of isolated thylakoid membranes are more complicated than those of intact leaves, and it should be carefully explained. Generally, uncouplers should be used for these experiments to evaluate the proper steady-state activity because the lumen acidification immediately suppresses the linear electron transport. This is not the case if ATPase sustains the intact activity but it is unlikely to happen in isolated thylakoid membranes. Unfortunately, I am not convinced that the authors succeeded to properly evaluate the O₂ photoreduction in isolated thylakoid membranes in the present system. Indeed, the O₂ evolution/consumption show transient behaves, which is likely to be dependent on the redox state of an electron donor for photosynthetic electron transport just before the illumination.

Line 152 and 176, In these subtitles, I found the same concern as mentioned above. PQ pool is a just pool. Which does give electrons to PQ pool? Theoretically, CEF neither consumes nor produces electrons. Finally, I understand that the authors measured the O₂ photoreduction with electrons already stored before the illumination in the photosynthetic electron transport system during the short light times until the electron transport is suppressed by lumen acidification. I am doubt that we can precisely discuss the molecular mechanisms of O₂ photoreduction based on such a limited result.

Line 210-212, I guess this is opposite. If PQ pool is strongly reduced, CEF cannot work well because there is less electron acceptor although it depends on which the limitation step is...

Line 227-234, In my knowledge, the acceptor-side limitation is not easy to evaluate only from P700 measurements. See the latest literatures (Theune et al. 2021; Furutani et al. 2022).

Figures 1 and 3, Why are the values shown in "nmol"? Please show them in the concentration e.g., "μM". Further, please calculate and show the O₂ evolution and consumption rate in μmol O₂ mg Chl-1 h⁻¹. This makes it much easier to compare the present results with previous reports.

References:

- Furutani R, Ohnishi M, Mori Y, Wada S, Miyake C (2022) The difficulty of estimating the electron transport rate at photosystem I. *J Plant Res* 135:565-577
- Hanawa H, Ishizaki K, Nohira K, Takagi D, Shimakawa G, Sejima T, Shaku K, Makino A, Miyake C (2017) Land plants drive photorespiration as higher electron-sink: comparative study of post-illumination transient O₂-uptake rates from liverworts to angiosperms through ferns and gymnosperms. *Physiol Plant* 161:138-149
- Ilík P, Pavlovič A, Kouřil R, Alboresi A, Morosinotto T, Allahverdiyeva Y, Aro E-M, Yamamoto H, Shikanai T (2017) Alternative electron transport mediated by flavodiiron proteins is operational in organisms from cyanobacteria up to gymnosperms. *New Phytol* 214:967-972
- Noridomi M, Nakamura S, Tsuyama M, Futamura N, Vladkova R (2017) Opposite domination of cyclic and pseudocyclic electron flows in short-illuminated dark-adapted leaves of angiosperms and gymnosperms. *Photosynth Res* 134:149-164
- Takagi D, Ishizaki K, Hanawa H, Mabuchi T, Shimakawa G, Yamamoto H, Miyake C (2017)

Diversity of strategies for escaping reactive oxygen species production within photosystem I among land plants: P700 oxidation system is prerequisite for alleviating photoinhibition in photosystem I. *Physiol Plant* 161:56-74

Theune ML, Hildebrandt S, Steffen-Heins A, Bilger W, Gutekunst K, Appel J (2021) In-vivo quantification of electron flow through photosystem I – Cyclic electron transport makes up about 35% in a cyanobacterium. *Biochim Biophys Acta Bioenerg* 1862:148353

Reviewers' comments are mentioned in **'bold'**, response to the reviewer is mentioned in 'black', the changes made in the main manuscript or in supplementary in response to the reviewers' comments are mentioned in **green** and the new line numbers after the changes are mentioned in **Red**.

Response to reviewer #1.

Comment 1:

Bag et. al present evidence that there is a strong photoprotective effect against damage by high light at low temperatures in Scots pine and to some extent also Norway spruce during early spring as compared to summer. This is reflected in a net O₂ consumption rather than an O₂ release. In their manuscript Bag et al interrogate the different points of the electron transport chain to determine where O₂ consumption occurs and ultimately where the electrons end in order to determine the mechanism at play. In the end, the conclusion is that none of the canonical protective mechanisms that typically are at play in plants display this effect. Instead, an increased expression of Flv proteins (similar to moss and cyanobacteria) is shown and by inference it is concluded that these act as an electron sink by reducing excess O₂.

The manuscript is clear and well written with a reasonable conclusion, albeit it is not hammered in with 7-inch nails. I have no major comments related to the science presented as it is all very sound. Overall, I found the manuscript a great read with some very nice and clear well explained experiments. Therefore, I think this manuscript would be a nice addition to the Nat. Comm. portfolio.

I do have a list of minor comments, that should be considered by the authors.

Authors reply:

We greatly appreciate the positive response.

Comment 2:

Line 37 – please introduce the shorthand ES already here.

Authors' reply:

Done (Line 37).

Comment 2:

Line 50 to 53 – This may seem a finicky comment, but these four lines had a large influence on my want to continue reading, so I will present the comment regardless of how small it may seem. As a reader, I am confused by the abrupt statement about plankton and immediate transition to boreal forests, where the relevance of these forests for O₂ production is defended. It would be better just to say that boreal forests contribute significantly to O₂ production due to their widespread presence in the northern regions.

Authors' reply:

Indeed, it is an abrupt jump. We have modified this part to the following:

Line 49-51: O₂ in the Earth's atmosphere is generated by photosynthetic organisms growing in water and on land. Boreal forests cover 14% of Earth's land (1.9 billion hectares) and account for 33% of Earth's total forests, thereby contributing significantly to the global carbon balance and O₂ production[1].

Comment 3:

Line 125 – Fig 1D seems to be a CTOE experiment not a MIMS experiment.

Authors' reply:

We have updated Figure 1 with new figure labelling (Line 838).

Comment 4:

Figure 1 – Is there a good reason for the hierarchy of labels in this figure? Otherwise, I suggest to change the labelling to be a – b – c in the first row and d – e – f in the second row or a – c – e in the first and b – d – f in the second. As it is now is confusing. – also very confusing that m/z 34 is shown to the left of m/z 32 (similar for Figure 2, that 32 is to the right of 34).

Authors' reply:

As mentioned above we have updated Fig 1 with new labelling. We have also changed the $^{16}\text{O}_2$ and $^{16,18}\text{O}_2$ axis in Fig 1B to match with Fig 1D, F (Line 838).

Comment 5:

Line 331 – In lune should be in line

Authors' reply:

Corrected.

Comment 6:

Line 440/441 – Do you mean to say that the samples contained 25% DMSO or was it perhaps a 100 mM stock

Authors' reply:

We meant to say 100mM DCMU stock, and it was prepared in DMSO. We have changed the sentence to the following,

Line 455-456: For inhibition of PSII activity, DCMU (3-(3,4-dichlorophenyl)-1,1-dimethylurea) (dissolved in DMSO) was added at a final concentration of 25 μM .

Comment 7:

Line 1215 – Provide = provided

Authors' reply:

Corrected.

Comment 8:

References citing Science papers have formatting issues – 3, 4 , 83

It might be an active choice, but you seem to have lost all superscripts and subscripts in your references and in some cases even have spaces inserted between H and 2 in H2 (ref. 5)

Authors' reply:

Corrected

Response to reviewer #2.

Comment 1 (we divided this comment in two sections 1.1 and 1.2 for easier response)

Comment 1.1:

Scots pine and Norway spruce show the strong O₂ consumption in early spring. By eliminating other possibilities, the authors claim that this high O₂ uptake depends on Flv. In fact, the Flv protein level increases in early spring. They observed the strong O₂ uptake even in the presence of DCMU. They suggest that the origin of electrons is the stromal pool, as well as PSII.

I understand that the genetical approach is not feasible in the non-model plants. However, the contribution of Flv to the photoprotection in the plants is not very surprising. The authors could not directly prove it in the process.

Authors' reply:

Indeed, we agree with the reviewer that the 'Flv activity' in conifers is not a completely new suggestion and we tried to make it clear in the manuscript. The novelty of our manuscript is not that we suggest that 'Flvs maybe active in conifers', rather, we substantiate the previous predictions with new and direct experimental data where their contribution is quantified under different conditions, and we could precisely show that in winter/early spring they play a major role in preventing photooxidative damage. We believe that our study provides thus far the strongest and most direct evidence for the action of Flvs in a non-model vascular plant, even if we could "not hammer it in with 7-inch nails" as reviewer 1 phrases it. We hope that the revisions in the current version will convince this reviewer that our manuscript has enough merit for publication in NatComm, specifically given the vast global importance of boreal forest and keeping in mind the genetic limitations that apply when studying these plants.

For directly validating the phenomena of 'O₂ consumption in winter/early spring' and involvement of Flavodiiron proteins in outdoor grown plants, we simulated winter/early spring like conditions in climate chambers and correlated the function and the abundance of the Flv protein when Pine seedling were adapted to freezing temperatures with moderately high light. To make these conclusions clearer, the paragraph 'The phenomenon could be reproduced in controlled condition' has been changed to the following,

Line 270-271: O₂ photoreduction is also triggered under artificially simulated semi-early spring conditions in the climate chambers.

Comment 1.2:

Another serious problem is the lack of evidence on the electron donor to the pathway. I guess that they consider the contribution of type II NDH. This is also likely, but the direct experimental evidence is missing especially because of the problem in Antimycin A (see below). The manuscript contains some interesting discoveries, but I have to say that it lacks the solid conclusion.

Authors' reply:

We fully acknowledge this issue, and the reviewer is correct that we consider the contribution of type II NDH as one of the major sources. This all relates to the complex issue about CEF and PCEF in conifers in winter/early spring. We provide below further arguments and data supporting our conclusion and point out how the manuscript has now been revised to substantiate this.

Could CEF be a source of electrons for O₂ photoreduction in early spring? As per our understanding, there are two ways of reducing PQ pool (Reducing PQ pool - electron donation to the PQ pool leading to formation of PQH₂); 1. Photochemically and 2. Non-photochemically (Sun et al. 2021). Photochemically PQ reduction is caused by H₂O oxidation in Photosystem II. Nonphotochemical reduction of PQ pool is mediated by Cyclic electron flow (PGR5/PGRL1) and metabolic oxidation of

organic molecules, such as NAD(P)H and NADH (the NDH pathway) (Braukmann, Kuzmina, and Stefanović 2009; Nawrocki et al. 2019; Wu et al. 2021). Please note that conifers lack typical NDH type I that is known for NDH mediated cyclic electron flow in angiosperms, hence, in case of conifers, only NDH type II pathway can be considered (Allorent et al. 2016; Braukmann, Kuzmina, and Stefanović 2009; Grebe et al. 2019; Wakasugi et al. 1994). When both H₂O oxidation and NDH-II pathway can generate electrons, CEF via PGR5/PGRL1 would neither generate nor consume electrons, only recycle the electrons that are already present (“injected” by H₂O oxidation and/or NDH pathway) in the intersystem pool - hence, it is named as cyclic electron flow (Alric 2015; Sun et al. 2021). Therefore, the true sources of electrons that could reduce the PQ pool in conifers would be either water oxidation and/or oxidation of NAD(P)H and/or NADH.

1. In experiments with DCMU (Fig 2A, B) we showed that, even though H₂O oxidation is blocked, O₂ consumption does not fade away, rather slightly increases; we explain this slight increase by the fact that the true O₂ consumption was masked by a minor O₂ production by H₂O oxidation in absence of DCMU in ES. Moreover, it is well established that PSII remains in a highly quenched state (Adams et al. 2004; Bag et al. 2020; Adam M Gilmore and Ball 2000; Ivanov et al. 2001, 2002; Krol et al. 2002), and therefore, donates significantly fewer electrons into the PQ pool that stem from photochemistry in PSII in ES compared to S. However, numerous reports suggested that the total electron pool in the PETC in winter is much larger than in summer (Ivanov et al. 2001, 2002; Savitch et al. 2010). This suggests that the electron donation is mediated by non-photochemical reduction of the PQ pool. This information is now mentioned in line 193-194.

2. We have now added measurements of the intersystem electron pool size ($e^-/P700$) on the intact needles as described previously (Asada, Heber, and Schreiber 1993). Briefly, intact needles were dark incubated for 20 min and then P700 oxidation was induced by far-red light for 5 min, in between a single turnover (ST) and a multi turnover (MT) flash was applied (Ivanov et al. 2001, 2002). Far red light preferentially excites PSI and therefore the contribution of PSII remains negligibility small, specifically in winter when PSII activity was reported to be severely restricted. The ratio of MT-area vs ST-area is represented as the intersystem e⁻ PQ pool (see the Figure above). This experiment indicated that the intersystem e⁻ pool was 3 times bigger in ES needles compared to S needles. This corroborates with the ratio of PQ/PQH₂, which showed approximately 2.5-3 times lower PQ/PQH₂ in ES compared to S, suggesting that the intersystem electrons were indeed donated to the PQ pool in the thylakoids (Fig 2C). This is also in line with previously reported values (Ivanov et al. 2001, 2002). Moreover, it was also reported by Ivanov et al. 2001 that along with intersystem electron pool, stromal e⁻ pool was also 5 times bigger and it was linked with NADPH/NADH mediated non photochemical reduction. From this, we conclude that indeed there were much more electrons in the chloroplast (since the measurement was done on needles) in ES than S. As linear electron flow was severely restricted in winter, these electrons were contributed non-photochemically to the PQ pool. We have added this figure in Fig 2C and discussed in the results (Line 186-187).

E. Dark reduction of PQ

3. We added measurements of the Post illumination rise of F_o' from intact needles, as an indicator of as an estimate of the dark reduction of the PQ pool by stromal reductants, as it has been previously reported in *Chlamydomonas* (Jans et al. 2008; Peltier, Aro, and Shikanai 2016) as a consequence of NDH II activity (Nda2). This post illumination rise of F_o' was also recently shown to be higher in presence of Flvs in algae and cyanobacteria compared to their Flv knockout mutants (Mattila et al. 2022). Similarly, we also found that F_o' rise was much higher in early spring pine needles compared to summer (please see the figure above). This information is now added in line 187-189 and in Fig 2D.

Note that the post illumination F_o' rise patterns were slightly different than previously reported in *Chlamydomonas sp* (Jans et al. 2008; Peltier, Aro, and Shikanai 2016). However, it was also recently shown that there remains species specific variation in post illumination F_o' rise (Mattila et al. 2022; Schuurmans et al. 2015), which could be related to the wavelength of the measuring light and/or excitation light for PSII and PSI. Moreover, under highly reducing environment, this constant rise of F_o' could also be an indication of constant electron donation to the PQ pool in dark.

Line 179-196 now reads:

Non-photochemically reduced PQ pool supplies electrons to PSI for the photoreduction of O_2

PQ, a common biological redox mediator, is present predominantly within the thylakoid lipid bilayer and plastoglobuli[37]. In higher plants, ~30% of the total PQ is photochemically active[38] and the rest is modulated via non-photochemical processes. O_2 consumption in presence of DCMU indicated that the electrons for photoreduction of O_2 came from an already stored e^- pool in the thylakoid, that could originate from non-photochemical sources in intact needles in absence of LEF during early spring.. Hence, we first measured the intersystem e^- pool size on intact needles, which suggested that the $e^-/P700$ was 3 times higher in ES compared to S (Fig 2C). Moreover, post illumination F_o' rise measurements suggested that non-photochemical dark reduction of the PQ pool indeed occurred in ES intact needles (Fig 2D). To decipher if our isolated thylakoid samples also have an increased non-photochemical electron pool, we analyzed the prenyl quinones by LC/QTOF mass spectrometry. In dark-adapted thylakoids, the PQ pool in ES thylakoids was predominantly reduced: the PQ/PQH2 ratio in ES was ~40% lower than in S (Fig. 2E). In addition, the ubiquinone pool (UQ) was heavily reduced (Supplementary Fig. S5). Interestingly, the stromal e^- pool was previously reported to be 5 times higher in ES pine needles than S[39].

At this moment we cannot pinpoint the source of electrons for the nonphotochemical reduction of the PQ pool. This is because, as mentioned, NAD(P)H reduction in chloroplast can be from several sources, such as glycolysis, malate shuttle, prenyl pool inter-conversion (UQ to MQ to PQ and vice versa) and NDH II activity. We simply do not have the means to decipher all of them. It would be interesting to know which pathway exactly donates e^- for non-photochemical reduction, but this is an open question that needs to be resolved by the future studies. However, we discuss the possibility of

NDH II activity for NAD(P)H reduction and non-photochemical reduction in context of previous reports, and have added to the discussion section:

Line 372-376: However, we found that in ES thylakoids, PQ remains in a massively reduced state (Fig. 2E) concomitant with 3 times higher intersystem e- pool (Fig 2C) and 5 times higher stromal e- pool in the intact pine needles[39], [55]. In absence or extremely low LEF, this non-photochemical reduction of the PQ (Fig 2D) could be linked to higher NAD(P)H reduction via chloroplastic NDH II activity as predicted previously[39], [51].

Concerning using Antimycin A as a CEF blocker: We recognise the work of Sugimoto et al., 2013, where *Pinus tedea* Pgr5 (gene that has been over expressed in *Arabidopsis*) show partial Antimycin A sensitivity and now the work is cited in the main text (Line 2225). Moreover, below we discuss three possibilities that may counter the notion of the reviewer's here.

1. Prg5 only cycles electrons and does not generate any (please also see the responses given above), therefore, CEF via PGR5/PRGL1 cannot be the true electron donor for O₂ photoreduction, irrespective of its lower sensitivity towards Antimycin A.
2. Based on results obtained by Sugimoto et al. (2013), we have used 3-fold higher concentration of Antimycin A, i.e., 30 μM compared to 10 μM in their study. Moreover, our intention was not excluding that PGR5 is involved, in fact, we mention (Line 389-391) that PGR5 could very well be involved in ATP generation in ES needles. This is also supported by the previous discovery that both ATP and ADP content was much higher in winter pine compared to summer (Ivanov et al. 2002). In winter/early spring, when the CBB cycle is blocked, higher ATP generation could be linked with higher PGR5 mediated CEF as respiration rates are also lower in freezing temperature.
3. CEF has been shown to be hindered in presence of a heavily reduced PQ pool (Alric 2015). Both our spectroscopic (Fig 2C) and biochemical data (Fig 2E) together with previous reports suggests that the PQ pool remains heavily reduced, hence, chances of CEF are small, at least in the beginning of illumination when we see strong O₂ consumption. Therefore, the electron donor to the PQ pool should be non-photochemical reduction that maybe contributed by NDH II. It is also noteworthy that discrimination between PGR5 and Flv pathway is nearly impossible without making genetic knockout(see Allahverdiyeva et al. 2011; Ilik et al. 2017; Jokel et al. 2018). It has been suggested recently that CEF (Pgr5) and PCEF (Flv) may work antagonistically under certain conditions (Dao et al. 2023).

Therefore, now we consider the possibility that Flvs first remove the excess electrons from the PQ pool upon illumination. This further can shift the equilibrium towards PQ from PQH₂ that may favour CEF to generate ATP, which is considered as a contribution from PSI to generate the *pmf* under LEF limited conditions (Jokel et al. 2018; Shikanai and Yamamoto 2017). Hence, now we discuss the possible synergistic play between Flv and CEF to maintain the redox balance in ES as below,

Line 388-395: However, as PGR5/PGRL1 was reported to be more abundant in ES than S[26], along with higher abundance of ATP in winter acclimated pine needles[55], hence, we speculate that PGR5-CEF may contribute to ATP production via proton gradient formation[59]. In ES when intersystem e- pool is much higher[28], [39], over reduction of the PQ could prevent CEF[41] and thereby hinder *pmf* generation[59] which could lower ATP production. But, electron scavenging by Flvs from the acceptor-side of PSI through O₂ photoreduction upon illumination could shift the PQ/PQH₂ balance and restore CEF. This would maintain the *pmf* and continue ATP generation even if LEF was restricted[22], [55].

Comment 2:

Figure S1. Yellow is hard to see.

Authors' reply:

We have now replaced yellow text in Supplementary figure S1 with dark brown.

Comment 3:

Briefly explain the essence of TR-MIMS assay in the main text. It would be helpful for readers if you explain that water was labelled.

Authors' reply:

We thank the reviewer for pointing this out. We had mentioned in Line 109-112 that H_2^{18}O was used for TR-MIMS assay in the first version of the manuscript. However, we agree that it will be helpful to provide more information and, therefore, have modified the sentence:

Line 110-114: Second, we performed time-resolved membrane-inlet mass spectrometry (TR-MIMS) assays with 10% of ^{18}O -enriched water (H_2^{18}O) (Supplementary Fig. S2) to discriminate between oxygen production (mainly $^{16}\text{O}^{18}\text{O}$) and consumption (mainly $^{16}\text{O}_2$) reactions, monitored at m/z 34 and m/z 32 signals, respectively[33].

Comment 4:

The text includes rather many abbreviations. I often had to go back the story to confirm some abbreviations. Is it necessary to use so many abbreviations, CTOE, MR, PR, PCEF, ...?

Authors' reply:

We apologise for this issue. Now we have replaced CTOE by Clark-electrode; MR by Mehler-reaction; PR by Photorespiration; PCEF by Pseudo-CEF.

Comment 5:

Explain green and blue lines in all figures.

Authors' reply:

Done. Now we have mentioned the green and blue lines at least once in every figure. However, since we keep the colour coding same for Summer and Early spring although the manuscript, therefore, we only show the legends with one graph within one figure.

Comment 6:

Ls124-130. This paragraph is hard to follow. Fig. 1B (L125)? in ES pine thylakoids (Ls128-129)?

Authors' reply:

We have now replaced this paragraph with:

Line 123-137.

These data indicate that, in vitro, the O_2 production of the S pine thylakoids is inhibited by a photoinactivation mechanism that is dependent on the O_2 level in the sample cuvette. Interestingly, spinach membranes in air-saturated buffer without PPBQ and FeCy supplementation did not show any O_2 exchange (Supplementary Fig. S1C), whereas ES and S pine thylakoids showed only O_2 consumption (Supplementary Fig. S3).

TR-MIMS data of spinach thylakoids were similar to the Clark-electrode measurements and both $^{16}\text{O}_2$ and $^{16,18}\text{O}_2$ exhibited similar kinetics (Fig. 1D). As the $^{16,18}\text{O}_2$ -traces at the beginning of illumination are much less sensitive to O_2 reduction (due to the initially low $^{16,18}\text{O}_2$ concentration in the measuring buffer), this suggests that in spinach thylakoids O_2 reduction is of minor importance. The spinach measurements also demonstrate that sufficient artificial electron acceptors have been added to sustain O_2 evolution for 60 s under our experimental conditions. In line with the Clark-electrode measurements, for pine S thylakoids (Fig. 1E, F, green line) only a transient O_2 production is observed for both oxygen

species, even at the reduced O₂ levels in the MIMS cell. In addition, also the O₂ consumption showed a transient behavior, possibly indicating a light-induced effect of PS I (see below)..

Comment 7:

L178. in the lumen of chloroplasts??? PQ is in the thylakoid membrane and PG. PQ in mitochondria?

Authors' reply:

We apologise for this error and have now updated the text.

Comment 8:

Figure 3B. It was reported that PGR5-dependent CET was resistant to Antimycin A in *Pinus sativa* (Sugimoto et al., 2013). Is it possible to confirm the amino acid sequence in your plants? At least, you have to mention the possibility that CET is resistant against Antimycin A. Without further confirmation, it is difficult to conclude anything on PGR5-dependent CET.

Authors' reply:

We thank the reviewer for pointing this out. We have now updated this result section (Line 220-229) and the discussion (Line 388-395). Unfortunately, there is no sequence available for *Pinus sylvestris* at this moment to the best of our knowledge. Please also see our response to comment 1.1 and 1.2.

We have updated the text regarding this issue as mentioned in response to comment 1.1 and 1.2. Moreover, we would like to point out that nowhere in the manuscript we disregard the fact that CEF is involved. Rather we have acknowledged the contribution of CEF is generating ATP in early spring when LEF and respiration is low. Now we have also discussed a possible interplay between Flv and CEF (PGR5 dependent) and Flvs shifting the PQ pool redox state (towards oxidised state) by consuming the excess electrons donated non-photochemically from the heavily reduced chloroplastic environment, that in turn could enhance CEF and promote ATP generation.

Line 388-395: However, as PGR5/PGRL1 was reported to be more abundant in ES than S[26], along with higher abundance of ATP in winter acclimated pine needles[55], hence, we speculate that PGR5-CEF may contribute to ATP production via proton gradient formation[59]. In ES when intersystem e-pool is much higher[28], [39], over reduction of the PQ could prevent CEF[41] and thereby hinder pmf generation[59] which could lower ATP production. But, electron scavenging by Flvs from the acceptor-side of PSI through O₂ photoreduction upon illumination could shift the PQ/PQH₂ balance and restore CEF. This would maintain the pmf and continue ATP generation even if LEF was restricted[22], [55].

Comment 9:

L231. What are luminal components. Does the plant have Cyt6? Do you consider other electron donors to P700?

Authors' reply:

We thank the reviewer for pointing out this phrasing mistake, and it has been corrected. To the best of our knowledge Pine and Spruce do not have Cyt6 in thylakoids, and at this moment we considered non photochemically reduced PQ as the electron donor to P700 as we do not know of any other donors in Pine and Spruce.

Comment 10:

L237. Does catalase function in chloroplasts in the plant?

Authors' reply:

Catalase has been removed.

Comment 11:

Figure 3E. Why did not you combine Figure 3E and S3? 35% of O₂ reduction is via MR in ES plants. You cannot ignore its contribution. In S plants, contribution of MR is really minor.

Authors' reply:

This is an excellent suggestion by the reviewer, we also had considered this possibility in the beginning, however, we decided against this, since we had to call the supplementary figure S3 in the very beginning of the results sections, where we show that without acceptor (PPBQ + FeCy) the trend of O₂ exchange is grossly negative and this is a contrasting light dependent gas exchange pattern in coniferous gymnosperms compared to angiosperms (Spinach). This observation is one of the key conclusions of the manuscript as well, hence, we wanted to call this figure very early in the text and keep it as Supplementary figure 3. However, if the reviewer/editor still want this to be in main figures, we can merge Fig S3 with the main Figure 1.

Upon illumination, at the PSI acceptor side, the Mehler reaction is activated which forms ROS that are scavenged by (per)oxidases. However, there are numerous sites, apart from the acceptor side of PSI, such as within PSII and Cyt b₆f, that also produces ROS which are scavenged by thylakoid bound ROS scavenging enzymes. Hence, we have now changed the sentence in the result section to consider the reviewers comment and we state that the Mehler-reaction only explains a fraction (Rather than minor) of the total O₂ consumption. As noted by the reviewer, compared to S, in ES the contribution of the Mehler-reaction is higher, hence, we also included the contribution of Mehler reaction in winter in the discussion section, as below,

Line 416-417: this does not mean other mechanisms for O₂-reduction are not involved in parallel, such as the Mehler-reaction (Fig 3E).

Comment 12:

L260. Smaller pines? Define the plant materials scientifically. Size, age, ...?

Authors' reply:

The requested detailed information is now provided in the Methods section under 'Plant material harvesting' (Line 440). We have also replaced the term 'smaller pine' in Line 273 with 'saplings'.

Comment 13:

PCEF via Flv. From this term I imagine the electron flow from water (generated by PSII) to water. But you suggest that the origin of electrons is mainly NADP. Is this PCET?

Authors' reply:

We agree that PCET in general is considered as electron flow from water (PSII) to water (via Flv from the acceptorside of PSI). Here, the source of electron is mainly non-photochemical in nature (NAD(P)H) and originates only partially from H₂O, at least in the early spring. Therefore, the term 'PCEF' could be misleading. This issue with the terminology is also raised by one of the other reviewers and it's been suggested to consider calling this phenomena as 'Mehler-like-reaction'. Therefore, we now call this pathway as Mehler-like reaction.

Comment 14:

Prg5/PrgI1 should be PGR5/PGRL1.

Authors' reply:

Corrected.

Comment 15:

L389. ndh-2 homolog? Do you mean the different enzyme from type II NDH?

Authors' reply:

At this moment we do not have the means to identify if type II NDH is a different form in conifers, hence, 'ndh-2 homolog' is removed from the text, and we simply call it ndh-2. To the best of our knowledge there are three different NDH-II in conifers; all of which are upregulated in the transcript level in spruce during winter months (Bag 2022; Bag et al. 2021) (please, see the figure below). Given the similarity between spruce and pine, the functional redundancy and the dominant O₂ photoreduction phenomena in both species, we predict, that these ndh-2 (that is known as functional NDH-II in algae chloroplast for non-photochemical reduction), is the best possible candidate for this. Some of the translated sequences of ndh-2 is also predicted to be localised in the chloroplast in spruce. Please see the expression data in the figure below,

Figure legend: Expression profile of three type II NDH genes present in *Picea abies*. Expression profiles were expressed in terms of variance stabilised counts. Open circles indicate each replicate, bars represent mean data point and error bars represent mean \pm SE ($n = 6-9$). Here S is considered as 3rd and 6th June samples and ES is considered as 22nd Feb and 27th March samples. Note that this part of the data was already published as a resource article in (Bag et al. 2021). The sequence of these genes was obtained from multiple transcript sequence alignment into a gene model.

Please note that we have not provided this figure in the main manuscript since this data was obtained during our seasonal gene expression study (Bag et al. 2021) and not from the same year needles that were subjected to gas exchange measurements in this manuscripts

Comment 16:

Ls396-398. What do you mean by primitive? Ref68 reported the exogenous Flv system in rice.

Authors' reply:

By the term 'primitive' we intended to express that the Flv pathway was previously characterised in lower organisms, that have emerged much earlier than gymnosperms and angiosperms. Since, this mechanism was also categorised as futile in (Ilić et al. 2017), we have changed structure of the sentence (Line 410-411).

Comment 17:

L407. I was really confused. Conifers do not contain type I NDH (photosynthetic complex I). Does your plant also lack type II NDH?

Authors' reply:

We have now corrected this mistake.

Comment 18:

There are many typos in the text.

Authors' reply:

We apologise for the typing errors and have corrected any typos that we could find.

Response to Reviewer #3:

Comment 1:

Bag et al. studied O₂ photoreduction in conifers. The photoreduction of O₂ mediated by Flv has already been reported by several recent literatures (Noridomi et al. 2017; Ilik et al. 2017; Takagi et al. 2017). The main novelties of this work are (1) quantitative evaluation of Flv-mediated O₂ photoreduction at the level of isolated thylakoid membranes and (2) seasonal difference of it. These new findings are worth being published in Nature Communications if they are properly verified. However, there are several concerns as follows:

Authors' reply:

We appreciate the positive response and thank the reviewer for the detailed recommendations and concerns that we address below.

Comment 2:

Line 55-73, Terminology for photosynthetic electron transport is strange. Flv can be categorized into "Mehler-like reaction". MR is also funny abbreviation for this. Pseudo-CEF is an often-used word, but I would suggest "Mehler-like reaction" is more suitable to Flv, because "pseudo-CEF" should be difficult to understand for non-expert.

Authors' reply:

Indeed, Flv pathway may be characterised as 'Mehler-like reaction'; thank you for this suggestion that we adopted now. Mehler reaction and Mehler-like reaction is no longer abbreviated in the text.

Comment 3:

Line 84, I agree with that the activity of photorespiration is lower in gymnosperms than in C3 plants, and further decreases at low temperature. However, certain activities have been measured by previous literatures e.g., (Hanawa et al. 2017), and the authors can mention this more.

Authors' reply:

Citations added in introduction (Line 82), and a sentence added to the discussion section:

Line 328-331: Among them, photorespiration in general remains lower in gymnosperms compared to angiosperms, specifically at low temperatures[29], moreover, it was easily excluded since we observed O₂ consumption in isolated thylakoid membranes.

Comment 4:

Line 88-91, This sentence is strange and should be clarified more. Normally, we cannot imagine that PSI uses the electrons much more than generated at PSII.

Authors' reply:

Modified, and references added to support the statement:

Line 88-89: we obtained clear evidence that O₂ photoreduction around PSI is much stronger than PSII O₂ evolution, as PSII remained extremely quenched in early spring[22].

Comment 5:

Line 103-130, This section needs to be seriously improved. The measurements of isolated thylakoid membranes are more complicated than those of intact leaves, and it should be carefully explained. Generally, uncouplers should be used for these experiments to evaluate the proper steady-state activity because the lumen acidification immediately suppresses the linear

electron transport. This is not the case if ATPase sustains the intact activity, but it is unlikely to happen in isolated thylakoid membranes. Unfortunately, I am not convinced that the authors succeeded to properly evaluate the O₂ photoreduction in isolated thylakoid membranes in the present system. Indeed, the O₂ evolution/consumption show transient behaviors, which is likely to be dependent on the redox state of an electron donor for photosynthetic electron transport just before the illumination.

Authors' reply:

O₂ exchange measurements on thylakoid membranes are well-established in our labs. The lack of most of the stromal components can be compensated by addition of artificial electron acceptors for photosystem II. We also point out that thylakoids were employed that are leaky to protons due to a previous freeze/thaw cycle (mentioned in Methods). Thus, there is no build-up of a proton gradient even if the ATPase is not working. The data with spinach thylakoids (Figure 1A, D), performed at the same Chl concentration, demonstrate that enough exogenous electron acceptors were added to sustain O₂ evolution over the whole illumination period, and that the freeze/thaw cycle is efficient in making the membranes leaky to protons. This information is now mentioned in Line 106-109.

Nevertheless, we agree with the reviewer that the transient behavior indicates some inhibition mechanism of PSII during the course of illumination. The time of inhibition appears to be O₂ dependent, thus could be due to some ROS formation under *in vitro* conditions, in line with the about 30% Mehler reaction identified by the catalase experiments (Fig 3E). Nevertheless, identifying the precise mechanism of inhibition is beyond this present study. To enhance the description of the data obtained, the presence of this inhibition is now specifically mentioned in the text. Importantly, the main conclusions of the manuscript regarding the differences between ES and S pine thylakoids gas exchange remain fully valid, as they can be clearly discerned from the initial 30 s of the O₂ exchange measurements.

We also note that the delay between illumination and O₂ signal in the beginning is due to the instrumental setup, now explained in the legend of Figure (1). We have modified the paragraph as follows:

Line 123-137:

These data indicate that, *in vitro*, the O₂ production of the S pine thylakoids is inhibited by a photoinactivation mechanism that is dependent on the O₂ level in the sample cuvette. Interestingly, spinach membranes in air-saturated buffer without PPBQ and FeCy supplementation did not show any O₂ exchange (Supplementary Fig. S1C), whereas ES and S pine thylakoids showed only O₂ consumption (Supplementary Fig. S3).

TR-MIMS data of spinach thylakoids were similar to the Clark-electrode measurements and both ¹⁶O₂ and ^{16,18}O₂ exhibited similar kinetics (Fig. 1D). As the ^{16,18}O₂-traces at the beginning of illumination are much less sensitive to O₂ reduction (due to the initially low ^{16,18}O₂ concentration in the measuring buffer), this suggests that in spinach thylakoids O₂ reduction is of minor importance. The spinach measurements also demonstrate that sufficient artificial electron acceptors have been added to sustain O₂ evolution for 60 s under our experimental conditions. In line with the Clark-electrode measurements, for pine S thylakoids (Fig. 1E, F, green line) only a transient O₂ production is observed for both oxygen species, even at the reduced O₂ levels in the MIMS cell. In addition, also the O₂ consumption showed a transient behavior, possibly indicating a light-induced effect of PS I (see below)..

Comment 6 (subdivided to 6.1 and 6.2 for easier response)

Comment 6.1:

Line 152 and 176, In these subtitles, I found the same concern as mentioned above.

Authors' reply:

Please see our response above which also applies here.

Comment 6.2:

PQ pool is a just pool. Which does give electrons to PQ pool? Theoretically, CEF neither consumes nor produces electrons. Finally, I understand that the authors measured the O₂ photoreduction with electrons already stored before the illumination in the photosynthetic electron transport system during the short light times until the electron transport is suppressed by lumen acidification. I am doubt that we can precisely discuss the molecular mechanisms of O₂ photoreduction based on such a limited result.

Authors' reply:

We fully agree with the analysis of the reviewer that under the invitro experimental conditions of the Clark-electrode and MIMS experiments the O₂ reduction is driven by electrons stored in the PQ pool prior to thylakoid isolation. Thus, the mechanism of the *in vivo* PQ pool reduction cannot be discerned from these experiments. **However, highly important and novel data regarding the size differences of the e⁻ pool in the PETC, as well as regarding the reduction mechanism of O₂ can be obtained.** As such, the data clearly indicate (i) a significantly larger electron pool for O₂ reduction in ES thylakoids as compared to S thylakoids, and (ii) in combination with catalase addition (Fig 3E), that the Mehler reaction accounts for only about 30% of O₂ reduction, indicating that another mechanism must exist (which we subsequently identify as Flv mediated O₂ reduction).

Indications that the PQ pool reduction is mainly due to non-photochemical i.e. metabolic processes, come from the following evidence. For photochemical reduction of PQ PSII must be active; however, in ES samples, PSII activity remains severely quenched (Bag et al. 2020; Gilmore and Ball 2000; Grebe et al. 2020; Ivanov et al. 2001, 2002). To further explore this, we now added measurement of intersystem e⁻ pool and post illuminatic rise of Fo' from intact needles (Fig 2C and E). The measured ratio of PQ/PQH₂ in the isolated thylakoid membranes (Fig 2D) also corroborated with the intact needle measurements. Which in turn, suggest that the already reduced PQ pool in the thylakoid measurements, indeed donate electrons for O₂ photoreduction and this reduction is caused by non-photochemical activities. The intersystem e⁻ pool size of intact needles was measured by P700 activity with single turnover (ST) and multi turnover flashes (MT) under FR light. This measurement preferentially activates PSI and therefore directly correlates with the internal electron pool in the PETC. In ES, PSII remained severely quenched, therefore, the contribution of PSII will be small and post illumination rise of Fo' suggested that the electron donor in PQ pool in ES is non-photochemical in nature and this is most likely linked to NDH-II activity. Hence, we added the following sentences in results (Line 179-196) and discussion section (Line 372-376).

Line 179-196 now reads:

Non-photochemically reduced PQ pool supplies electrons to PSI for the photoreduction of O₂

PQ, a common biological redox mediator, is present predominantly within the thylakoid lipid bilayer and plastoglobuli[37]. In higher plants, ~30% of the total PQ is photochemically active[38] and the rest is modulated via non-photochemical processes. O₂ consumption in presence of DCMU indicated that the electrons for photoreduction of O₂ came from an already stored e⁻ pool in the thylakoid, that could originate from non-photochemical sources in intact needles in absence of LEF during early spring. Hence, we first measured the intersystem e⁻ pool size on intact needles, which suggested that the e⁻/P700 was 3 times higher in ES compared to S (Fig 2C). Moreover, post illumination Fo' rise suggested that non-photochemical dark reduction of the PQ pool indeed occurred in ES intact needles (Fig 2D). To decipher if our isolated thylakoid samples also had an increased non-photochemical electron pool, we analyzed the prenyl quinones by LC/QTOF mass spectrometry. In dark-adapted thylakoids, the PQ pool in ES thylakoids was predominantly reduced: the PQ/PQH₂ ratio in ES was ~40% lower than in S (Fig. 2E). In addition, the ubiquinone pool (UQ) was heavily reduced (Supplementary Fig. S5). Interestingly, the stromal e⁻ pool was previously reported to be 5 times higher in ES pine needles than S[39].

Line 372-376: However, we found that in ES thylakoids, PQ remains in a massively reduced state (Fig. 2E) concomitant with 3 times higher intersystem e⁻ pool (Fig 2C) and 5 times higher stromal e⁻ pool in the intact pine needles[39], [55]. In absence or extremely low LEF, this non-photochemical reduction of the PQ (Fig 2D) could be linked to higher NAD(P)H reduction via chloroplastic NDH II activity as predicted previously[39], [51].

Comment 7:

Line210-212, I guess this is opposite. If PQ pool is strongly reduced, CEF cannot work well because there is less electron acceptor although it depends on which the limitation step is...

Author's reply:

Indeed, now we have modified this to:

Line 221-229: While higher P700 oxidation could in principle be due to CEF[26], recent reports showed that a reduced intersystem PQ pool (Fig. 2C, E) hinders CEF[41]. To completely block CEF we added Antimycin A (CEF blocker)[42]. The morphology of pine needles [43] and their wax coating may reduce the penetration of Antimycin A into the needles. Nevertheless, it was recently reported that PGR5 in *Pinus teda* is partially sensitive to Antimycin A[44]. Therefore, we employed 3 times higher Antimycin A concentration than reported previously for monitoring the P700 kinetics in the thylakoid membranes by the SP method. While Antimycin A induced some differences, we found a similar overall trend for thylakoids with Antimycin A compared to without Antimycin A addition measurements on intact needles, *i.e.*, with increasing irradiance, Y(ND) was higher, and Y(NA) was lower in ES than S (compare Figs 3A and 3B).

Comment 8:

Line 227-234, In my knowledge, the acceptor-side limitation is not easy to evaluate only from P700 measurements. See the latest literatures (Theune et al. 2021; Furutani et al. 2022).

Author's reply:

We thank the reviewer for pointing this out and providing these literatures. Specifically for this reason we performed both SP methods (Determination of conventional Y(NA) and Y(ND)) and then without flash measured steady state P700 kinetics with only FR light. In steady state P700 remained oxidised (Fig 3D) when electrons were supplied from by PQH₂ (Fig 2C-E). And it's well established that in winter Calvin cycle remains blocked, therefore the electrons must be taken up from the acceptor-side. It's also noteworthy that higher PSI oxidation is previously been measured as well. Now we have discussed this more in context of these references in the results section adding:

Line 231-235: Recently SP method was shown not to be a reliable measure for a possible acceptor-side limitation of PSI as the steady-state concentration of P700⁺ may be influenced by electrons contributed from the PSII via the PQ [45], [46]. Therefore, to determine steady-state P700⁺, we exposed S and ES thylakoid membranes to six intermittent cycles of far-red (FR) light[47] in the presence of antimycin A (blocks CEF) (Fig. 3C).

Comment 9:

Figures 1 and 3, Why are the values shown in "nmol"? Please show them in the concentration e.g., "μM". Further, please calculate and show the O₂ evolution and consumption rate in μmol O₂ mg Chl-1 h-1. This makes it much easier to compare the present results with previous reports.

Author's reply:

We apologize for this mistake, we meant nmol/ml. Now as per the reviewer's suggestion, we have changed nmol/ml to μmol/L, which is denoted as μM in the main figures.

We have added the rates in the supplementary figure S11 - where all the rates have been calculated as the slope from first 10 seconds of O₂ exchange and expressed in terms of per mg Chl per hour. This

was done to capture the maximum contrast of O₂ exchange in S and ES thylakoid membranes (please also see the last section of our response to comment 5). We have provided details of this rate calculation in the figure legend.

Comment 10:

References:

Furutani R, Ohnishi M, Mori Y, Wada S, Miyake C (2022) The difficulty of estimating the electron transport rate at photosystem I. J Plant Res 135:565-577

Hanawa H, Ishizaki K, Nohira K, Takagi D, Shimakawa G, Sejima T, Shaku K, Makino A, Miyake C (2017) Land plants drive photorespiration as higher electron-sink: comparative study of post-illumination transient O₂-uptake rates from liverworts to angiosperms through ferns and gymnosperms. Physiol Plant 161:138-149

Ilík P, Pavlovič A, Kouřil R, Alboresi A, Morosinotto T, Allahverdiyeva Y, Aro E-M, Yamamoto H, Shikanai T (2017) Alternative electron transport mediated by flavodiiron proteins is operational in organisms from cyanobacteria up to gymnosperms. New Phytol 214:967-972

Noridomi M, Nakamura S, Tsuyama M, Futamura N, Vladkova R (2017) Opposite domination of cyclic and pseudocyclic electron flows in short-illuminated dark-adapted leaves of angiosperms and gymnosperms. Photosynth Res 134:149-164

Takagi D, Ishizaki K, Hanawa H, Mabuchi T, Shimakawa G, Yamamoto H, Miyake C (2017) Diversity of strategies for escaping reactive oxygen species production within photosystem I among land plants: P700 oxidation system is prerequisite for alleviating photoinhibition in photosystem I. Physiol Plant 161:56-74

Theune ML, Hildebrandt S, Steffen-Heins A, Bilger W, Gutekunst K, Appel J (2021) In-vivo quantification of electron flow through photosystem I – Cyclic electron transport makes up about 35% in a cyanobacterium. Biochim Biophys Acta Bioenerg 1862:148353

Author's reply:

We thank the reviewer for providing these references and we have now added/updated all of them.

Reviewers' Comments:

Reviewer #1:

Remarks to the Author:

I think the corrections made by the authors address the comments well. I am still of the opinion that this is a great paper.

Note that in ref 71, the authors name is all capitals for some reason.

Reviewer #2:

Remarks to the Author:

In Scots pine and Norway spruce, high level of O₂ reduction was observed in early spring. This was accompanied by the increase in the Flv level. This O₂ reduction was partly insensitive to DCMU but the origin of electrons is unclear. This would be the summary of the revised version. I could not find any fair reason to change my initial evaluation in this version especially because the manuscript lacks the definitive conclusion. However, I would be happy to see this manuscript even in a good journal of plant physiology.

The revised version still contains several problems in science.

- 1) Figure 2C and E shows that the PQ pool was more reduced in ES plants than in S plants. Electrons were already in the intersystem, probably in the PQ pool, before the measurement. The results do not suggest the non-photochemical reduction of the PQ pool.
- 2) Figure 2D. The post-illumination increase in chlorophyll fluorescence is influenced by many factors. Especially, it is influenced by the recovery of quenched state (F_o') when the intensity of actinic light is high (320 μmol photons m⁻² s⁻¹). It should be confirmed that it really monitored the PQ reduction by applying FR light, etc. Notably, it specifically monitors the PQ reduction depending on the NDH complex which is absent in conifers.
- 3) Ls224-225. Ref44 reports that PGR5 of the pine is resistant to antimycin A. It was not partially sensitive. The conclusion was based on the study on the Arabidopsis plants expressing the pine PGR5. The result is not related to the penetration of antimycin A.
- 4) L124. What do you mean by the photoinactivation mechanism?

Reviewer #3:

Remarks to the Author:

The authors well responded to the comments that I previously suggested. Now, the manuscript has been improved enough.

Reviewers' comments are mentioned in **'bold'**, response to the reviewer is mentioned in 'black', the changes made in the main manuscript or in supplementary in response to the reviewers' comments are mentioned in **Green** and the new line numbers after the changes are mentioned in **Red**. Please note that this time we separated the main text from the supplementary as per Nature communication guidelines, hence, the ref numbers may change.

Response to reviewer #1.

Comment 1:

I think the corrections made by the authors address the comments well. I am still of the opinion that this is a great paper.

Note that in ref 71, the authors name is all capitals for some reason.

Author's reply:

We appreciate the positive response and have now corrected error in the reference.

Response to reviewer #2:

Comment 1:

In Scots pine and Norway spruce, high level of O₂ reduction was observed in early spring. This was accompanied by the increase in the Flv level. This O₂ reduction was partly insensitive to DCMU but the origin of electrons is unclear. This would be the summary of the revised version. I could not find any fair reason to change my initial evaluation in this version especially because the manuscript lacks the definitive conclusion. However, I would be happy to see this manuscript even in a good journal of plant physiology.

Author's reply:

We thank the reviewer for his/her/their constructive criticism on the 2nd version of our manuscript. Please find our response to your comments below.

Comment 2:

The revised version still contains several problems in science.

Figure 2C and E shows that the PQ pool was more reduced in ES plants than in S plants. Electrons were already in the intersystem, probably in the PQ pool, before the measurement. The results do not suggest the non-photochemical reduction of the PQ pool.

Author's reply:

We agree with the reviewer that our data in the first instance show that the PQ pool was highly reduced in both intact needles (Fig 2C) and thylakoids (Fig 2E). The question is: from where came the electrons for PQ reduction?

Here, the reviewer is not convinced that the PQ pool was reduced non-photochemically. We like to emphasize that there are several lines of evidence that rule out the only other alternative – photochemical reduction. Therefore, to carefully address the reviewer's concerns we added data from additional experiments on both intact needles and isolated thylakoids from the exact samples that has been used in the main manuscript for representation of the figures (samples from 2017-2020, these measurement were already done on fresh needles during those year). These new data, our previous evidence and data of the literature can be summarized as follows:

2.1. Firstly, there is already evidence in the literature that photochemical linear electron flow in winter/early spring conifer needles is extremely low or absent (Ivanov et al. 2001; 2002; Sveshnikov et al. 2006; Grebe et al. 2020; Adam M Gilmore and Ball 2000; A M Gilmore et al. 2003). We have also ourselves shown this in our previous pine manuscript in Fig 1e and f (Bag et al. 2020) by measuring PSII fluorescence in vivo on intact pine needles. This low PSII LEF was seen over three consecutive years (2018, 2017, 2016) (Bag et al. 2020) and therefore, was a consistent early spring phenomena. In the figure below we also show the quantification of PSII activity in another

independent way, from P700 measurements with the application of single turnover flash (ST) on a FR light background from intact needles. Measuring of PSII activity in terms of P700 oxidation via ST depends on the concept that in far red light (FR) PSI is preferentially excited [We are aware that

FR may also activate PSII to a minor extent in some species (Houille-Vernes et al. 2011), but this is in a much smaller extent in early spring in pine (Krol et al. 2002; Ivanov et al. 2001)]. Application of a ST white light flash on top of FR light, allows activation and completion of one PSII turnover, hence, represents a reliable and accurate measure of the functional PSII complexes in vivo [For detailed discussion see (Losciale et al. 2008)]. The area under the curve of the ST was approx. 9 times larger in S compared to ES, similar to the PSII water oxidation capacity noted in Fig 1F. This shows that very few electrons are generated photochemically under these, quite extreme winter/early spring conditions, regardless the method employed for the measurement of PSII activity.

In theory it could be argued that even though the PSII fluorescence (Bag et al. 2020) and H₂O oxidation is low (Fig 1c, d, e and f in current manuscript), if the processes in the downstream of PSII, i.e., PSI activity, is lower than PSII activity, then the low PSII activity in early spring needles could still fill up the intersystem e⁻ pool. However, we have shown previously in Fig 2a in Bag et al. Nat Comm, 2020 (Bag et al. 2020) that PSI activity was in fact higher in early spring than PSII activity. This information has now been emphasized in the introduction Line 89-91 and in results Line 186.

Line 90 now reads,

we obtained clear evidence that O₂ photoreduction around PSI is much stronger than PSII-related O₂ evolution, as PSII remained extremely quenched and PSI activity was higher than PSII in early spring[22]....

Line 186 now reads,

that could originate from non-photochemical sources in intact needles in absence of LEF during early spring[22].

2.2. We now also performed P700 re-reduction measurements after FR-light illumination on intact needles and compared with isolated thylakoid membranes (see the figure below). For this, first, on intact needles, we induced P700 oxidation via FR-light for 4-5 minutes, then switched off the FR-light and followed the re-reduction (Supplementary Fig S5a) and calculated the time constant [Tau (s)] of the decay by using a mono exponential decay fit (Supplementary Fig S5b), as was fitted previously in (Wood et al. 2018; Sveshnikov et al. 2006). Under FR illumination PSI was preferentially excited, and thereby PSI drew electrons from the PQ pool that were present in the intersystem prior to the illumination (Note that LEF was severely restricted so activation of PSII by FR-light would negligibly small). With time, under FR-light illumination, PSI kept draining electrons from the intersystem and reached a steady state (after 30 s to 1 min). FR-light was kept on for further 2-3 minutes to ensure that all the intersystem electrons were completely drained. Hence, this would put all PSI reaction centres in an oxidised state (P700⁺). Since this measurement was performed on intact needles, all stromal components were intact, so, later when the FR-light is switched off, electrons could be injected into the PQ pool non-photochemically (via PGR5-CEF and other stromal reductant pathways). This caused reduction of the P700⁺ to form P700. As the

electrons were donated non-photochemically, the rate of P700 re-reduction was proportional to the electron flow through combination of different non-photochemical pathways. Supplementary Fig S5a, b suggested that the P700+ re-reduction rate was approx. 5 times faster in ES needles (Tau = 0.5 s) compared to S (Tau = 2.1 s).

2.3. We also performed the same P700 re-reduction measurement on isolated thylakoid membranes (from the same batch of needles as above), but this time with 40-45 s FR-illumination with higher FR intensities. Shorter time with higher intensity was still enough to reach P700+ stationary phase (as noted in intact needle measurement) but minimised any potential ROS formation in PSI that could have occurred with longer illumination in absence of stromal fraction. As thylakoid membranes lack stromal fraction, the P700 re-reduction in this measurement was purely dependent on two factors 1. non-photochemical electron donation by PGR5-CEF and 2. already stored electrons in the PQ pool prior to the illumination (If FR-light could not drain all electrons). In this measurement (Supplementary Fig S5c, d) we found the Tau of P700 re-reduction was much slower in both S (17.4 s) and ES samples (11.4 s) compared to intact needle measurements. The time constant changed approx. 8 times in S thylakoid and approx. 25 times in ES thylakoid compared to the respective intact needles. Therefore, from these two measurements (Supplementary Fig S5b, d), we could clearly demonstrate that in ES much more (3 times) electrons came from stromal reductants (Such as NDH type II, glycolysis, malate shuttle, interconversion between PQ, MQ and UQ) and not from PSII.

Taken together, we are fully convinced that the majority of the electrons in the intersystem was indeed originated from non-photochemical sources. To clear this, we have now added the citation for low LEF in early spring in line 186 on intact pine needles and modified the text there in (Line 190-192) and added the below data with the supplementary figure S5. We also note that the other two reviewers seem to be agree with us in this issue.

Line 190-193 now reads,

This was also confirmed by comparing the changes in the P700 re-reduction rate upon switching off FR-light illumination, in intact needles with thylakoid samples (Supplementary Fig. S5a-d).

Comment 3:

Figure 2D. The post-illumination increase in chlorophyll fluorescence is influenced by many factors. Especially, it is influenced by the recovery of the quenched state (Fo') when the intensity of actinic light in high (320 μmol photons m⁻² s⁻¹). It should be confirmed that it really monitored the PQ reduction by applying FR light, etc. Notably, it specifically monitors the PQ reduction depending on the NDH complex which is absent in conifers.

Author's reply:

This is a legitimate question. Below we present three arguments in support of the choice our actinic light for this measurement in Fig 2D,

- 3.1. 320 μmol of photons $\text{m}^{-2} \text{s}^{-1}$ light is not high for this material (outdoor grown pine trees) which may have already faced 1000 μmol of photons $\text{m}^{-2} \text{s}^{-1}$ in sub-zero temperatures the day before sampling [see Fig 1a,b and c in (Bag et al. 2020)], where this point has been established and,
- 3.2. In an earlier study [Ivanov et al, 2001; Planta, Fig 7c, d (Ivanov et al. 2002)], the post illumination rise of F_o' fluorescence was measured with 350 μmol photons $\text{m}^{-2} \text{s}^{-1}$ in the same species that is under investigation here (*Pinus sylvestris*). For better comparability and reproducibility with earlier literature the same method and similar light intensity were employed in the present manuscript.
- 3.3. Finally, this year (2023) we collected new “ES state” samples, which was possible right now since the weather conditions in Umeå are still below zero (down to -15°C at dawn) and solar radiation was 300–400 watt m^{-2} (600-1000 moles of photons $\text{m}^{-2} \text{s}^{-1}$), F_v/F_m in the needles are 0.35-0.45. On newly sampled needles we performed the post illumination rise of F_o' in FR-light background and compared with the F_o' rise in dark background as done previously by (Houille-Vernes et al. 2011) (See figure below). Pre-illumination of the intact needles was done by 320 μmol of photons $\text{m}^{-2} \text{s}^{-1}$ actinic light for 5-6 mins. Upon switching off the AL, in dark, F_o' kept rising as seen previously in

Figure legend: Post illumination rise of fluorescence (F_o') in dark and FR-light background measured on intact ES needles. AL light intensity for pre-illumination was 320 μmoles of photon $\text{m}^{-2} \text{s}^{-1}$ for 5-6 mins. In this measurement needles from 5 pine trees (The same trees that were used for sample harvesting in the rest of the manuscript) were pulled together. Data represents average of four traces from independent four measurements.

the ES samples, confirming that the same phenomena (as in Fig 2D, 2017-2020 samples) occurred also this year (2022-2023 season), but with a slightly smaller amplitude than previous years. When post AL illumination rise was measured in the presence of FR-light, the F_o' rise was diminished. This clearly demonstrated that the chosen AL intensity for this measurement did not cause any quenching, hence, the F_o' rise was indeed due to non-photochemical electron donation to the PQ pool. We do not add this data in the manuscript as the measurement year is different (2023) compared to the other figures (2017-2020) (hence, the magnitude of dark F_o' rise is different compared to Fig 2D and may confuse readers) and also the method is well-established for pine.

Therefore, to address the reviewer's concern, we now added the following sentences in the methods section in Line 537-542,

After 20 minutes of dark adaptation, the intact needles were subjected to a constant actinic red light (AL) of 320 μmol of photons $\text{m}^{-2} \text{s}^{-1}$ illumination for 5-6 minutes until steady state level of F_s was reached as reported previously in intact pine needles [53]. Then the actinic light was switched off and the transients from F_s to F_o' were recorded for 120 s. Note that, post illumination rise of F_o' is not influenced due to induction of artificial quenching by AL light intensity [22] in early spring.

Comment 4:

Ls224-225. Ref44 reports that PGR5 of the pine is resistant to antimycin A. It was not partially sensitive. The conclusion was based on the study on the Arabidopsis plants expressing the pine PGR5. The result is not related to the penetration of antimycin A.

Author's reply:

We feel that there may be some misunderstanding in the terminology used in ref44 (now ref45) and would like to cite a sentence from the Abstract of Sugimoto's manuscript: "We discovered that ferredoxin-dependent plastoquinone reduction in ruptured chloroplasts was less sensitive to antimycin

A in *Arabidopsis* that over-accumulated PGR5 (PROTON GRADIENT REGULATION 5) originating from *Pinus taeda* (PtPGR5) than that in the wild type." Similar statements were also used throughout the same manuscript. To the best of our knowledge, the authors of never claimed that "PGR5 of the pine is resistant to antimycin A", but it was rather "less sensitive to antimycin A". Therefore, we respectfully disagree with the reviewer and believe that the reviewers' statement "...PGR5 of the pine is resistant to antimycin A." based on ref 44 could be a case of oversimplification. We would like to give the reasons why we strongly believe the above,

4.1. It is shown in Yang et al. (2020) that a clear effect of Antimycin A (AA) could be observed in *Pinus sylvestris* needles, and AA was seen lowering the PSI yield suggesting blockage of PRG5-CEF in early spring [See Fig 5d and the corresponding result section in (Yang et al. 2020)]. In both spruce and pine, AA was shown to be perfectly capable of blocking of PGR5-CEF. Hence, the statement 'pine PGR5 is Antimycin A resistant' appears to be conditionally correct and could be a result of experimental limitation in the reference 44. Now, why do we think the that the effect of AA noted in reference 44 could be a limitation? See point 4.3 below.

4.2. With both information at hand, i.e., indirect evidence of PGR5 AA resistance in ref 45 (Sugimoto et al. 2013) and more recent direct evidence of PGR5 AA sensitivity in Pine (Yang et al. 2020), we validated the effect of Antimycin A on our own thylakoid samples before its application. For this, we again measured P700 re-reduction on the thylakoids, but this time added 30 μM AA prior to the measurement (Supplementary Fig S5e, f) and compared the results with no AA addition as shown in supplementary figure S5c, d. With AA addition to the thylakoids, the Tau of P700 re-reduction became much slower in both S and ES samples (Supplementary Fig S5e, f) compared to no AA addition (Supplementary Fig S5c, d). This clearly suggested that AA in our hand, could easily block electron flow through PGR5 and therefore, slow down the P700 re-reduction. It is noteworthy that two more conclusions could also be drawn from the above experiment,

- Even after PGR5-CEF was blocked AA, the re-reduction of P700 in ES thylakoids still faster (Tau = 19.2 s) than S thylakoids (Tau = 25.8 s). This corroborated with the notion that indeed the intersystem e- pool was much bigger in ES than S.
- Tau of P700 re-reduction slowed by approximately 8 s in both S and ES. This suggested that the contribution of electrons in the PQ pool by CEF was probably similar in both S and ES. However, since, PSI population differs among S and ES samples, the rate of electron donation through CEF-PSI⁻¹ (per PSI) could still be higher.

4.3. In the reference 45, authors over-expressed *Pinus taeda* (not *Pinus sylvestris*) PGR5 under a constitutive 35s promoter in *Arabidopsis thaliana*, however, Yang et al. 2020 clearly demonstrated effect of AA on *Pinus sylvestris*. There are possibilities that could explain why the data of Sugimoto et al. 2013 may not correctly reflect the situation in the intact pine needles. With a lot of current knowledge of conifer genomes (Nystedt et al. 2013; Bag et al. 2021), now we know that in conifers there could be multiple transcripts from one gene (Usually, number of transcripts per gene is much higher in conifers compared to *Arabidopsis*). It is possible that the PtPGR5 protein that was expressed in *Arabidopsis* may be different from the native pine PGR5 protein. Another reason could be also that the AA effect on Pine PGR5 could be species dependent. Hence, expressing the pine PGR5 protein in *Arabidopsis* (as in ref45) may have caused some placebo effect.

4.4. It is also possible that it the concentration of AA required for blocking pine PRG5 is higher; Yang et al, 2020 observed that the AA effect on pine PGR5 required a higher concentration (200 μM). So,

instead of performing intact needle experiments with very high concentration of AA, we performed the experiments on isolated thylakoid membranes and added only a slightly higher concentration (30 μM), than in ref 44 (10 μM).

We now explain this more carefully in the text in Line 225-233.

Line 225-233 now reads,

Therefore, to completely block CEF we used Antimycin A (known CEF blocker)[42]. One earlier report suggested that PGR5 in *Pinus tecta* is resistant to Antimycin A43 (at 10 μM) when it was expressed in *Arabidopsis*. However, a very recent report clearly demonstrated that Antimycin A effectively blocks the PGR5-CEF pathway in intact needles of *Pinus sylvestris* (at 200 μM)[26]. Hence, we used 3 times higher Antimycin A concentration than reported previously[43] and at this concentration, Antimycin A was effective in modulating P700 re-reduction kinetics (Supplementary Fig. S5c-f).

Comment 5:

L124. What do you mean by the photoinactivation mechanism?

Author's reply:

There are still contrasting ideas in the field concerning photoinactivation mechanisms We think a detailed discussion on this is outside the scope of this paper but have added a short statement on Line 125.

References:

- Bag, Pushan, Volha Chukhutsina, Zishan Zhang, Suman Paul, Alexander G Ivanov, Tatyana Shutova, Roberta Croce, Alfred R Holzwarth, and Stefan Jansson. 2020. "Direct Energy Transfer from Photosystem II to Photosystem I Confers Winter Sustainability in Scots Pine." *Nature Communications* 11 (1): 6388. <https://doi.org/10.1038/s41467-020-20137-9>.
- Bag, Pushan, Jenna Lihavainen, Nicolas Delhomme, Thomas Riquelme, Kathryn M Robinson, and Stefan Jansson. 2021. "An Atlas of the Norway Spruce Needle Seasonal Transcriptome." *The Plant Journal*. <https://doi.org/https://doi.org/10.1111/tpj.15530>.
- Gilmore, A M, S Matsubara, M C Ball, D H Barker, and S Itoh. 2003. "Excitation Energy Flow at 77 K in the Photosynthetic Apparatus of Overwintering Evergreens." *Plant, Cell & Environment* 26 (7): 1021–34.
- Gilmore, Adam M, and Marilyn C Ball. 2000. "Protection and Storage of Chlorophyll in Overwintering Evergreens." *Proceedings of the National Academy of Sciences* 97 (20): 11098–101.
- Grebe, Steffen, Andrea Trotta, Azfar Ali Bajwa, Ilaria Mancini, Pushan Bag, Stefan Jansson, Mikko Tikkanen, and Eva-Mari Aro. 2020. "Specific Thylakoid Protein Phosphorylations Are Prerequisites for Overwintering of Norway Spruce (*Picea Abies*) Photosynthesis." *Proceedings of the National Academy of Sciences* 117 (30): 17499–509.
- Houille-Vernes, Laura, Fabrice Rappaport, Francis-André Wollman, Jean Alric, and Xenie Johnson. 2011. "Plastid Terminal Oxidase 2 (PTOX2) Is the Major Oxidase Involved in Chlororespiration in *Chlamydomonas*." *Proceedings of the National Academy of Sciences* 108 (51): 20820–25.
- Ivanov, A, P Sane, Y Zeinalov, Gardeström Malmberg, Per Gardeström, N Huner, and G Öquist. 2001. "Photosynthetic Electron Transport Adjustments in Overwintering Scots Pine (*Pinus Sylvestris* L.)." *Planta* 213 (4): 575–85.
- Ivanov, A, P Sane, Y Zeinalov, I Simidjiev, N Huner, and G Öquist. 2002. "Seasonal Responses of Photosynthetic Electron Transport in Scots Pine (*Pinus Sylvestris* L.) Studied by Thermoluminescence." *Planta* 215 (3): 457–65.
- Krol, Marianna, Vaughan M Hurry, Denis P Maxwell, Lada Malek, Alexander G Ivanov, and Norman P A Huner. 2002. "Low Growth Temperature Inhibition of Photosynthesis in Cotyledons of Jack Pine Seedlings (*Pinus Banksiana*) Is Due to Impaired Chloroplast Development." *Canadian Journal of Botany* 80 (10): 1042–51.
- Losciale, Pasquale, Riichi Oguchi, Luke Hendrickson, Alexander B Hope, Luca Corelli-Grappadelli, and Wah Soon Chow. 2008. "A Rapid, Whole-tissue Determination of the Functional Fraction of PSII after Photoinhibition of Leaves Based on Flash-induced P700 Redox Kinetics." *Physiologia Plantarum* 132 (1): 23–32.

- Nystedt, Björn, Nathaniel R Street, Anna Wetterbom, Andrea Zuccolo, Yao-Cheng Lin, Douglas G Scofield, Francesco Vezzi, Nicolas Delhomme, Stefania Giacomello, and Andrey Alexeyenko. 2013. "The Norway Spruce Genome Sequence and Conifer Genome Evolution." *Nature* 497 (7451): 579–84.
- Sugimoto, Kazuhiko, Yuki Okegawa, Akihiko Tohri, Terri A Long, Sarah F Covert, Toru Hisabori, and Toshiharu Shikanai. 2013. "A Single Amino Acid Alteration in PGR5 Confers Resistance to Antimycin A in Cyclic Electron Transport around PSI." *Plant and Cell Physiology* 54 (9): 1525–34.
- Sveshnikov, Dmitry, Ingo Ensminger, Alexander G Ivanov, Douglas Campbell, Jon Lloyd, Christiane Funk, Norman P A Hüner, and Gunnar Öquist. 2006. "Excitation Energy Partitioning and Quenching during Cold Acclimation in Scots Pine." *Tree Physiology* 26 (3): 325–36.
- Wood, William H J, Craig MacGregor-Chatwin, Samuel F H Barnett, Guy E Mayneord, Xia Huang, Jamie K Hobbs, C Neil Hunter, and Matthew P Johnson. 2018. "Dynamic Thylakoid Stacking Regulates the Balance between Linear and Cyclic Photosynthetic Electron Transfer." *Nature Plants* 4 (2): 116–27.
- Yang, Qi, Nicolás E Blanco, Carmen Hermida-Carrera, Nóra Lehotai, Vaughan Hurry, and Åsa Strand. 2020. "Two Dominant Boreal Conifers Use Contrasting Mechanisms to Reactivate Photosynthesis in the Spring." *Nature Communications* 11 (1): 1–12.

Response to reviewer #3:

The authors well responded to the comments that I previously suggested. Now, the manuscript has been improved enough.

Author's reply:

We are delighted that the reviewer finds this modified version improved.

Reviewers' Comments:

Reviewer #2:

Remarks to the Author:

The authors tried to respond to my concerns. I was satisfied by the effort.